# MICo: Improved representations via sampling-based state similarity for Markov decision processes

**Pablo Samuel Castro**[*]
Google Research, Brain Team

**Tyler Kastner**[*]
McGill University

**Prakash Panangaden**
McGill University

**Mark Rowland**
DeepMind

## Abstract

We present a new behavioural distance over the state space of a Markov decision process, and demonstrate the use of this distance as an effective means of shaping the learnt representations of deep reinforcement learning agents. While existing notions of state similarity are typically difficult to learn at scale due to high computational cost and lack of sample-based algorithms, our newly-proposed distance addresses both of these issues. In addition to providing detailed theoretical analysis, we provide empirical evidence that learning this distance alongside the value function yields structured and informative representations, including strong results on the Arcade Learning Environment benchmark.

## 1 Introduction

The success of reinforcement learning (RL) algorithms in large-scale, complex tasks depends on forming useful representations of the environment with which the algorithms interact. Feature selection and feature learning has long been an important subdomain of RL, and with the advent of deep reinforcement learning there has been much recent interest in understanding and improving the representations learnt by RL agents.

Much of the work in representation learning has taken place from the perspective of *auxiliary tasks* [Jaderberg et al., 2017, Bellemare et al., 2017, Fedus et al., 2019]; in addition to the primary reinforcement learning task, the agent may attempt to predict and control additional aspects of the environment. Auxiliary tasks shape the agent's representation of the environment *implicitly*, typically via gradient descent on the additional learning objectives. As such, while auxiliary tasks continue to play an important role in improving the performance of deep RL algorithms, our understanding of the effects of auxiliary tasks on representations in RL is still in its infancy.

In contrast to the implicit representation shaping of auxiliary tasks, a separate line of work on *behavioural metrics*, such as bisimulation metrics [Desharnais et al., 1999, 2004, Ferns et al., 2004, 2006], aims to capture structure in the environment by learning a metric measuring behavioral similarity between states. Recent works have successfully used behavioural metrics to shape the representations of deep RL agents [Gelada et al., 2019, Zhang et al., 2021, Agarwal et al., 2021a]. However, in practice behavioural metrics are difficult to estimate from both statistical and computational perspectives, and these works either rely on specific assumptions about transition dynamics to make the estimation tractable, and as such can only be applied to limited classes of environments, or are applied to more general classes of environments not covered by theoretical guarantees.

The principal objective of this work is to develop new measures of behavioral similarity that avoid the statistical and computational difficulties described above, and simultaneously capture richer information about the environment. We introduce the *MICo (**M**atching under **I**ndependent **Co**uplings) distance*, and develop the theory around its computation and estimation, making comparisons with existing metrics on the basis of computational and statistical efficiency. We demonstrate the usefulness

---

[*]Equal contribution. Correspondence to Pablo Samuel Castro: psc@google.com.

35th Conference on Neural Information Processing Systems (NeurIPS 2021).

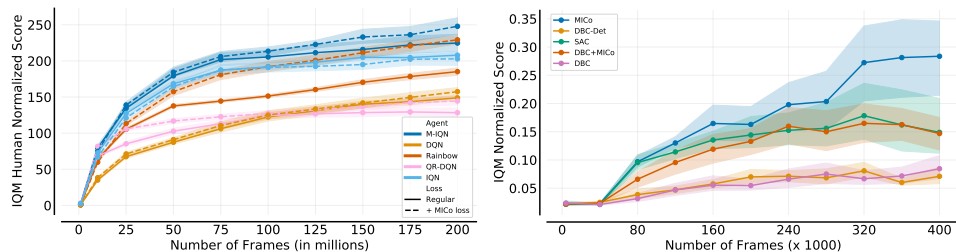

Figure 1: Interquantile Mean human normalized scores of all the agents and losses on the ALE suite (left) and on the DM-Control suite (right), both run with five independent seeds for each agent and environment. In both suites MICo provides a clear advantage.

of the representations that MICo yields, both through empirical evaluations in small problems (where we can compute them exactly) as well as in two large benchmark suites: (1) the Arcade Learning Environment [Bellemare et al., 2013, Machado et al., 2018], in which the performance of a wide variety of existing value-based deep RL agents is improved by directly shaping representations via the MICo distance (see Figure 1, left), and (2) the DM-Control suite [Tassa et al., 2018], in which we demonstrate it can improve the performance of both Soft Actor-Critic [Haarnoja et al., 2018] and the recently introduced DBC [Zhang et al., 2021] (see Figure 1, right).

## 2 Background

Before describing the details of our contributions, we give a brief overview of the required background in reinforcement learning and bisimulation. We provide more extensive background in Appendix A.

**Reinforcement learning.** We consider a Markov decision process $(\mathcal{X}, \mathcal{A}, \gamma, P, r)$ defined by a finite state space $\mathcal{X}$, finite action space $\mathcal{A}$, transition kernel $P : \mathcal{X} \times \mathcal{A} \to \mathscr{P}(\mathcal{X})$, reward function $r : \mathcal{X} \times \mathcal{A} \to \mathbb{R}$, and discount factor $\gamma \in [0, 1)$. For notational convenience we will write $P_x^a$ and $r_x^a$ for transitions and rewards, respectively. Policies are mappings from states to distributions over actions: $\pi \in \mathscr{P}(\mathcal{A})^{\mathcal{X}}$ and induce a *value function* $V^\pi : \mathcal{X} \to \mathbb{R}$ defined via the recurrence: $V^\pi(x) := \mathbb{E}_{a \sim \pi(x)} \left[ r_x^a + \gamma \mathbb{E}_{x' \sim P_x^a}[V^\pi(x')] \right]$. In RL we are concerned with finding the optimal policy $\pi^* = \arg\max_{\pi \in \mathscr{P}(\mathcal{A})^{\mathcal{X}}} V^\pi$ from interaction with sample trajectories with an MDP, *without knowledge of $P$ or $r$* (and sometimes not even $\mathcal{X}$), and the optimal value function $V^*$ induced by $\pi^*$.

**State similarity and bisimulation metrics.** Various notions of similarity between states in MDPs have been considered in the RL literature, with applications in policy transfer, state aggregation, and representation learning. The *bisimulation metric* [Ferns et al., 2004] is of particular relevance for this paper, and defines state similarity in an MDP by declaring two states $x, y \in \mathcal{X}$ to be close if their immediate rewards are similar, and the transition dynamics at each state leads to next states which are also judged to be similar. This self-referential notion is mathematically formalised by defining the bisimulation metric $d^\sim$ as the unique fixed-point of the operator $T_K : \mathcal{M}(X) \to \mathcal{M}(X)$, where $\mathcal{M}(X) = \{d \in [0, \infty)^{\mathcal{X} \times \mathcal{X}} : d \text{ symmetric and satisfies the triangle inequality}\}$ is the set of pseudometrics on $\mathcal{X}$, given by $T_K(d)(x, y) = \max_{a \in \mathcal{A}}[|r_x^a - r_y^a| + \gamma W_d(P_x^a, P_y^a)]$. Here, $W_d$ is the Kantorovich distance (also known as the Wasserstein distance) over the set of distributions $\mathscr{P}(\mathcal{X})$ with base distance $d$, defined by $W_d(\mu, \nu) = \inf_{X \sim \mu, Y \sim \nu} \mathbb{E}[d(X, Y)]$, for all $\mu, \nu \in \mathscr{P}(\mathcal{X})$, where the infimum is taken over all couplings of $(X, Y)$ with the prescribed marginals [Villani, 2008].

The mapping $T_K$ is a $\gamma$-contraction on $\mathcal{M}(X)$ under the $L^\infty$ norm [Ferns et al., 2011], and thus by standard contraction mapping arguments analogous to those used to study value iteration, it has a unique fixed point, the bisimulation metric $d^\sim$. Ferns et al. [2004] show that this metric bounds differences in the optimal value function, hence its importance in RL:

$$|V^*(x) - V^*(y)| \le d^\sim(x, y) \quad \forall x, y \in \mathcal{X}. \tag{1}$$

**Representation learning in RL.** In large-scale environments, it is infeasible to express value functions directly as vectors in $\mathbb{R}^{\mathcal{X} \times \mathcal{A}}$. Instead, RL agents must approximate value functions in a more concise manner, by forming a *representation* of the environment, that is, a feature embedding $\phi : \mathcal{X} \to \mathbb{R}^M$, and predicting state-action values linearly from these features. *Representation learning* is the problem of finding a useful representation $\phi$. Increasingly, deep RL agents are equipped with

additional losses to aid representation learning. A common approach is to require the agent to make additional predictions (so-called *auxilliary tasks*) with its representation, typically with the aid of extra network parameters, with the intuition that an agent is more likely to learn useful features if it is required to solve many related tasks. We refer to such methods as *implicit* representation shaping, since improved representations are a side-effect of learning to solve auxiliary tasks.

Since bisimulation metrics capture additional information about the MDP in addition to that summarised in value functions, bisimulation metrics are a natural candidate for auxiliary tasks in deep reinforcement learning. Gelada et al. [2019], Agarwal et al. [2021a], and Zhang et al. [2021] introduce auxiliary tasks based on bisimulation metrics, but require additional assumptions on the underlying MDP in order for the metric to be learnt correctly (Lipschitz continuity, deterministic, and Gaussian transitions, respectively). The success of these approaches provides motivation in this paper to introduce a notion of state similarity applicable to arbitrary MDPs, without further restriction. Further, we learn this state similarity *explicitly*: that is, without the aid of any additional network parameters.

## 3 Advantages and limitations of the bisimulation metric

The bisimulation metric $d^\sim$ is a strong notion of distance on the state space of an MDP; it is useful in policy transfer through its bound on optimal value functions [Castro and Precup, 2010] and because it is so stringent, it gives good guarantees for state aggregations [Ferns et al., 2004, Li et al., 2006]. However, it has been difficult to use at scale and compute online, for a variety of reasons that we summarize below.

(i) **Computational complexity.** The metric can be computed via fixed-point iteration since the operator $T_K$ is a contraction mapping. The map $T_K$ contracts at rate $\gamma$ with respect to the $L^\infty$ norm on $\mathcal{M}$, and therefore obtaining an $\varepsilon$-approximation of $d^\sim$ under this norm requires $O(\log(1/\varepsilon)/\log(1/\gamma))$ applications of $T_K$ to an initial pseudometric $d_0$. The cost of each application of $T_K$ is dominated by the computation of $|\mathcal{X}|^2|\mathcal{A}| W_d$ distances for distributions over $\mathcal{X}$, each costing $\tilde{O}(|\mathcal{X}|^{2.5})$ in theory [Lee and Sidford, 2014], and $\tilde{O}(|\mathcal{X}|^3)$ in practice [Pele and Werman, 2009, Guo et al., 2020a, Peyré and Cuturi, 2019]. Thus, the overall practical cost is $\tilde{O}(|\mathcal{X}|^5|\mathcal{A}|\log(\varepsilon)/\log(\gamma))$.

(ii) **Bias under sampled transitions.** Computing $T_K$ requires access to the transition probability distributions $P_x^a$ for each $(x, a) \in \mathcal{X} \times \mathcal{A}$ which, as mentioned in Section 2, are typically not available; instead, stochastic approximations to the operator of interest are employed. Whilst there has been work in studying online, sample-based approximate computation of the bisimulation metric [Ferns et al., 2006, Comanici et al., 2012], these methods are generally biased, in contrast to sample-based estimation of standard RL operators.

(iii) **Lack of connection to non-optimal policies.** One of the principal behavioural characterisations of the bisimulation metric $d^\sim$ is the upper bound shown in Equation (1). However, in general we do not have $|V^\pi(x) - V^\pi(y)| \leq d^\sim(x, y)$ for arbitrary policies $\pi \in \Pi$; a simple example is illustrated in Figure 2. More generally, notions of state similarity that the bisimulation metric encodes may not be closely related to behavioural similarity under an arbitrary policy $\pi$. Thus, learning about $d^\sim$ may not in itself be useful for large-scale reinforcement learning agents.

Property (i) expresses the intrinsic computational difficulty of computing this metric. Property (ii) illustrates the problems associated with attempting to move from operator-based computation to online, sampled-based computation of the metric (for example, when the environment dynamics are unknown). Finally, property (iii) shows that even if the metric is computable exactly, the information it yields about the MDP may not be practically useful. Although $\pi$-bisimulation (introduced by Castro

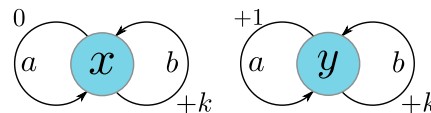

Figure 2: MDP illustrating that the upper bound for any $\pi$ is not generally satisfied. Here, $d^\sim(x, y) = (1 - \gamma)^{-1}$, but for $\pi(b|x) = 1, \pi(a|y) = 1$, we have $|V^\pi(x) - V^\pi(y)| = k(1 - \gamma)^{-1}$.

[2020] and extended by Zhang et al. [2021]) addresses property (iii), their practical algorithms are limited to MDPs with deterministic transitions [Castro, 2020] or MDPs with Gaussian transition kernels [Zhang et al., 2021]. Taken together, these three properties motivate the search for a metric without these shortcomings, which can be used in combination with deep reinforcement learning.

# 4 The MICo distance

We now present a new notion of distance for state similarity, which we refer to as *MICo* (**M**atching under **I**ndependent **Co**uplings), designed to overcome the drawbacks described above.

Motivated by the drawbacks described in Section 3, we make several modifications to the operator $T_K$ introduced above: (i) in order to deal with the prohibitive cost of computing the Kantorovich distance, which optimizes over all coupling of the distributions $P_x^a$ and $P_y^a$, we use the independent coupling; (ii) to deal with lack of connection to non-optimal policies, we consider an on-policy variant of the metric, pertaining to a chosen policy $\pi \in \mathscr{P}(\mathcal{A})^\mathcal{X}$. This leads us to the following definition.

**Definition 4.1.** Given $\pi \in \mathscr{P}(\mathcal{A})^\mathcal{X}$, the *MICo update operator* $T_M^\pi : \mathbb{R}^{\mathcal{X} \times \mathcal{X}} \to \mathbb{R}^{\mathcal{X} \times \mathcal{X}}$ is:

$$(T_M^\pi U)(x,y) = |r_x^\pi - r_y^\pi| + \gamma \mathbb{E}_{\substack{x' \sim P_x^\pi \\ y' \sim P_y^\pi}} [U(x',y')] \tag{2}$$

for all $U : \mathcal{X} \times \mathcal{X} \to \mathbb{R}$, with $r_x^\pi = \sum_{a \in \mathcal{A}} \pi(a|x) r_x^a$ and $P_x^\pi = \sum_{a \in \mathcal{A}} \pi(a|x) P_x^a(\cdot)$ for all $x \in \mathcal{X}$.

As with the bisimulation operator, this can be thought of as encoding desired properties of a notion of similarity between states in a self-referential manner; the similarity of two states $x, y \in \mathcal{X}$ should be determined by the similarity of the rewards and the similarity of the states they lead to.

**Proposition 4.2.** The operator $T_M^\pi$ is a contraction mapping on $\mathbb{R}^{\mathcal{X} \times \mathcal{X}}$ with respect to the $L^\infty$ norm.

*Proof.* See Appendix B. $\qquad\square$

The following corollary now follows immediately from Banach's fixed-point theorem and the completeness of $\mathbb{R}^{\mathcal{X} \times \mathcal{X}}$ under the $L^\infty$ norm.

**Corollary 4.3.** The MICo operator $T_M^\pi$ has a unique fixed point $U^\pi \in \mathbb{R}^{\mathcal{X} \times \mathcal{X}}$, and repeated application of $T_M^\pi$ to any initial function $U \in \mathbb{R}^{\mathcal{X} \times \mathcal{X}}$ converges to $U^\pi$.

Having defined a new operator, and shown that it has a corresponding fixed-point, there are two questions to address: Does this new notion of distance overcome the drawbacks of the bisimulation metric described above; and what does this new object tell us about the underlying MDP?

## 4.1 Addressing the drawbacks of the bisimulation metric

We introduced the MICo distance as a means of overcoming some of the shortcomings associated with the bisimulation metric, described in Section 3. In this section, we provide a series of results that show that the newly-defined notion of distance addressess each of these shortcomings. The proofs of these results rely on the following lemma, connecting the MICo operator to a lifted MDP. This result is crucial for much of the analysis that follows, so we describe the proof in full detail.

**Lemma 4.4 (Lifted MDP).** The MICo operator $T_M^\pi$ is the Bellman evaluation operator for an auxiliary MDP.

*Proof.* Given the MDP specified by the tuple $(\mathcal{X}, \mathcal{A}, P, R)$, we construct an auxiliary MDP $(\widetilde{\mathcal{X}}, \widetilde{\mathcal{A}}, \widetilde{P}, \widetilde{R})$, by taking the state space to be $\widetilde{\mathcal{X}} = \mathcal{X}^2$, the action space to be $\widetilde{\mathcal{A}} = \mathcal{A}^2$, the transition dynamics to be given by $\widetilde{P}_{(u,v)}^{(a,b)}((x,y)) = P_u^a(x) P_v^b(y)$ for all $(x,y), (u,v) \in \mathcal{X}^2$, $a, b \in \mathcal{A}$, and the action-independent rewards to be $\widetilde{R}_{(x,y)} = |r_x^\pi - r_y^\pi|$ for all $x, y \in \mathcal{X}$. The Bellman evaluation operator $\widetilde{T}^{\widetilde{\pi}}$ for this auxiliary MDP at discount rate $\gamma$ under the policy $\widetilde{\pi}(a,b|x,y) = \pi(a|x)\pi(b|y)$ is given by (for all $U \in \mathbb{R}^{\mathcal{X} \times \mathcal{X}}$ and $(x,y) \in \mathcal{X} \times \mathcal{X}$):

$$(\widetilde{T}^{\widetilde{\pi}} U)(x,y) = \widetilde{R}_{(x,y)} + \gamma \sum_{(x',y') \in \mathcal{X}^2} \widetilde{P}_{(x,y)}^{(a,b)}((x',y')) \widetilde{\pi}(a,b|x,y) U(x',y')$$

$$= |r_x^\pi - r_y^\pi| + \gamma \sum_{(x',y') \in \mathcal{X}^2} P_x^\pi(x') P_y^\pi(y') U(x',y') = (T_M^\pi U)(x,y). \qquad\square$$

**Remark 4.5.** Ferns and Precup [2014] noted that the bisimulation metric can be interpreted as the optimal value function in a related MDP, and that the functional $T_K$ of $T_K$ can be interpreted as a Bellman optimality operator. However, their proof was non-constructive, the related MDP being characterised via the solution of an optimal transport problem. In contrast, the connection described

above is constructive, and will be useful in understanding many of the theoretical properties of MICo. Ferns and Precup [2014] also note that the $W_d$ distance in the definition of $T_K$ can be upper-bounded by taking a restricted class of couplings of the transition distributions. The MICo metric can be viewed as restricting the coupling class precisely to the singleton containing the independent coupling.

With Lemma 4.4 established, we can now address each of the points (i), (ii), and (iii) from Section 3.

**(i) Computational complexity.** The key result regarding the computational complexity of computing the MICo distance is as follows.

**Proposition 4.6** (**MICo computational complexity**). The computational complexity of computing an $\varepsilon$-approximation in $L^\infty$ to the MICo metric is $O(|\mathcal{X}|^4 \log(\varepsilon)/\log(\gamma))$.

*Proof.* Since, by Proposition 4.2, the operator $T_M^\pi$ is a $\gamma$-contraction under $L^\infty$, we require $\mathcal{O}(\log(\varepsilon)/\log(\gamma))$ applications of the operator to obtain an $\varepsilon$-approximation in $L^\infty$. Each iteration of value iteration updates $|\mathcal{X}|^2$ table entries, and the cost of each update is $\mathcal{O}(|\mathcal{X}|^2)$, leading to an overall cost of $O(|\mathcal{X}|^4 \log(\varepsilon)/\log(\gamma))$. □

In contrast to the bisimulation metric, this represents a computational saving of $O(|\mathcal{X}|)$, which arises from the lack of a need to solve optimal transport problems over the state space in computing the MICo distance. There is a further saving of $\mathcal{O}(|\mathcal{A}|)$ that arises since MICo focuses on an individual policy $\pi$, and so does not require the max over actions in the bisimulation operator definition.

**(ii) Online approximation.** Due to the interpretation of the MICo operator $T_M^\pi$ as the Bellman evaluation operator in an auxiliary MDP, established in Lemma 4.4, algorithms and associated proofs of correctness for computing the MICo distance online can be straightforwardly derived from standard online algorithms for policy evaluation. We describe a straightforward approach, based on the TD(0) algorithm, and also note that the wide range of online policy evaluation methods incorporating off-policy corrections and multi-step returns, as well as techniques for applying such methods at scale, may also be used.

Given a current estimate $U_t$ of the fixed point of $T_M^\pi$ and a pair of observations $(x, a, r, x')$, $(y, b, \tilde{r}, y')$ generated under $\pi$, we can define a new estimate $U_{t+1}$ via

$$U_{t+1}(x, y) \leftarrow (1 - \epsilon_t(x, y))U_t(x, y) + \epsilon_t(x, y)(|r - \tilde{r}| + \gamma U_t(x', y')) \tag{3}$$

and $U_{t+1}(\tilde{x}, \tilde{y}) = U_t(\tilde{x}, \tilde{y})$ for all other state-pairs $(\tilde{x}, \tilde{y}) \neq (x, y)$, for some sequence of stepsizes $\{\epsilon_t(x, y) \mid t \geq 0, (x, y) \in \mathcal{X}^2\}$. Sufficient conditions for convergence of this algorithm can be deduced straightforwardly from corresponding conditions for TD(0). We state one such result below. An important caveat is that the correctness of this particular algorithm depends on rewards depending only on state; one can switch to state-action metrics if this hypothesis is not satisfied.

**Proposition 4.7.** Suppose rewards depend only on state, and consider the sequence of estimates $(U_t)_{t \geq 0}$, with $U_0$ initialised arbitrarily, and $U_{t+1}$ updated from $U_t$ via a pair of transitions $(x_t, a_t, r_t, x'_t)$, $(y_t, b_t, \tilde{r}_t, y'_t)$ as in Equation (3). If all state-pairs tuples are updated infinitely often, and stepsizes for these updates satisfy the Robbins-Monro conditions. Then $U_t \to U^\pi$ almost surely.

*Proof.* Under the assumptions of the proposition, the update described is exactly a TD(0) update in the lifted MDP described in Lemma 4.4. We can therefore appeal to Proposition 4.5 of Bertsekas and Tsitsiklis [1996] to obtain the result. □

Thus, in contrast to the Kantorovich metric, convergence to the exact MICo metric is possible with an online algorithm that uses sampled transitions.

**(iii) Relationship to underlying policy.** In contrast to the bisimulation metric, we have the following on-policy guarantee for the MICo metric.

**Proposition 4.8.** For any $\pi \in \mathscr{P}(\mathcal{A})^{\mathcal{X}}$ and states $x, y \in \mathcal{X}$, we have $|V^\pi(x) - V^\pi(y)| \leq U^\pi(x, y)$.

*Proof.* We apply a coinductive argument [Kozen, 2007] to show that if $|V^\pi(x) - V^\pi(y)| \leq U(x, y)$ for all $x, y \in \mathcal{X}$, for some $U \in \mathbb{R}^{\mathcal{X} \times \mathcal{X}}$ symmetric in its two arguments, then we also have $|V^\pi(x) - V^\pi(y)| \leq (T_M^\pi U)(x, y)$ for all $x, y \in \mathcal{X}$. Since the hypothesis holds for the constant function $U(x, y) = 2 \max_{z, a} |r(z, a)|/(1 - \gamma)$, and $T_M^\pi$ contracts around $U^\pi$, the conclusion then

follows. Therefore, suppose the hypothesis holds. Then we have

$$V^\pi(x) - V^\pi(y) = r_x^\pi - r_y^\pi + \gamma \sum_{x' \in \mathcal{X}} P_x^\pi(x')V(x') - \gamma \sum_{y' \in \mathcal{X}} P_y^\pi(y')V(y')$$

$$\leq |r_x^\pi - r_y^\pi| + \gamma \sum_{x',y' \in \mathcal{X}} P_x^\pi(x')P_y^\pi(y')(V^\pi(x') - V^\pi(y'))$$

$$\leq |r_x^\pi - r_y^\pi| + \gamma \sum_{x',y' \in \mathcal{X}} P_x^\pi(x')P_y^\pi(y')U(x',y') = (T_M^\pi U)(x,y) \,.$$

By symmetry, $V^\pi(y) - V^\pi(x) \leq (T_M^\pi U)(x,y)$, as required. $\qquad\square$

## 4.2 Diffuse metrics

To characterize the nature of the fixed point $U^\pi$, we introduce the notion of a *diffuse metric*.

**Definition 4.9.** Given a set $\mathcal{X}$, a function $d : \mathcal{X} \times \mathcal{X} \to \mathbb{R}$ is a *diffuse metric* if the following axioms hold: (i) $d(x,y) \geq 0$ for any $x, y \in \mathcal{X}$; (ii) $d(x,y) = d(y,x)$ for any $x, y \in \mathcal{X}$; (iii) $d(x,y) \leq d(x,z) + d(y,z) \; \forall x, y, z \in \mathcal{X}$.

These differ from the standard metric axioms in the first point: we no longer require that a point has zero self-distance, and two distinct points may have zero distance. Notions of this kind are increasingly common in machine learning as researchers develop more computationally tractable versions of distances, as with entropy-regularised optimal transport distances [Cuturi, 2013], which also do not satisfy the axiom of zero self-distance.

An example of a diffuse metric is the Łukaszyk–Karmowski distance [Łukaszyk, 2004], which is used in the MICo metric as the operator between the next-state distributions. Given a diffuse metric space $(\mathcal{X}, \rho)$, the Łukaszyk–Karmowski distance $d_{\mathrm{LK}}^\rho$ is a diffuse metric on probability measures on $\mathcal{X}$ given by $d_{\mathrm{LK}}^\rho(\mu, \nu) = \mathbb{E}_{x \sim \mu, y \sim \nu}[\rho(x,y)]$. This example demonstrates the origin of the name *diffuse* metrics: the non-zero self distances arises from a point being spread across a probability distribution. In terms of the Łukaszyk–Karmowski distance, the MICo distance can be written as the fixed point $U^\pi(x,y) = |r_x^\pi - r_y^\pi| + d_{\mathrm{LK}}(U^\pi)(P_x^\pi, P_y^\pi)$. This characterisation leads to the following result.

**Proposition 4.10.** The MICo distance is a diffuse metric.

*Proof.* Non-negativity and symmetry of $U^\pi$ are clear, so it remains to check the triangle inequality. To do this, we define a sequence of iterates $(U_k)_{k \geq 0}$ in $\mathbb{R}^{\mathcal{X} \times \mathcal{X}}$ by $U_0(x,y) = 0$ for all $x, y \in \mathcal{X}$, and $U_{k+1} = T_M^\pi U_k$ for each $k \geq 0$. Recall that by Corollary 4.3 that $U_k \to U^\pi$. We will show that each $U_k$ satisfies the triangle inequality by induction. By taking limits on either side of the inequality, we will then recover that $U^\pi$ itself satisfies the triangle inequality. The base case of the inductive argument is clear from the choice of $U_0$. For the inductive step, assume that for some $k \geq 0$, $U_k(x,y) \leq U_k(x,z) + U_k(z,y)$ for all $x, y, z \in \mathcal{X}$. Now for any $x, y, z \in \mathcal{X}$, we have

$$U_{k+1}(x,y) = |r_x^\pi - r_y^\pi| + \gamma \mathbb{E}_{X' \sim P_x^\pi, Y' \sim P_y^\pi}[U_k(X',Y')]$$

$$\leq |r_x^\pi - r_z^\pi| + |r_z^\pi - r_y^\pi| + \gamma \mathbb{E}_{X' \sim P_x^\pi, Y' \sim P_y^\pi, Z' \sim P_z^\pi}[U_k(X',Z') + U_k(Z',Y')]$$

$$= U_{k+1}(x,z) + U_{k+1}(z,y) \,. \qquad\square$$

It is interesting to note that a state $x \in \mathcal{X}$ has zero self-distance iff the Markov chain induced by $\pi$ initialised at $x$ is deterministic, and the magnitude of a state's self-distance is indicative of the amount of "dispersion" in the distribution. Hence, in general, we have $U^\pi(x,x) > 0$, and $U^\pi(x,x) \neq U^\pi(y,y)$ for distinct states $x, y \in \mathcal{X}$. See the appendix for further discussion of diffuse metrics and related constructions.

## 5 The MICo loss

The impetus of our work is the development of principled mechanisms for directly shaping the representations used by RL agents so as to improve their learning. In this section we present a novel loss based on the MICo update operator $T_M^\pi$ given in Equation (2) that can be incorporated into any RL agent. Given the fact that MICo is a diffuse metric that can admit non-zero self-distances, special

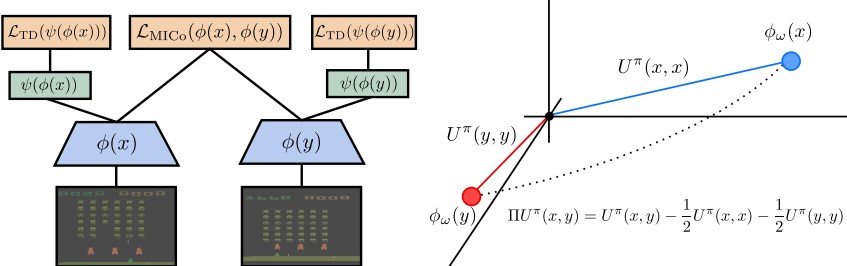

Figure 3: **Left:** Illustration of network architecture for learning MICo; **Right:** The projection of MICo distances onto representation space.

care needs to be taken in how these distances are learnt; indeed, traditional mechanisms for measuring distances between representations (e.g. Euclidean and cosine distances) are geometrically-based and enforce zero self-distances.

We assume an RL agent learning an estimate $Q_{\xi,\omega}$ defined by the composition of two function approximators $\psi$ and $\phi$ with parameters $\xi$ and $\omega$, respectively: $Q_{\xi,\omega}(x,\cdot) = \psi_\xi(\phi_\omega(x))$ (note that this can be the critic in an actor-critic algorithm such as SAC). We will refer to $\phi_\omega(x)$ as the *representation* of state $x$ and aim to make distances between representations match the MICo distance; we refer to $\psi_\xi$ as the *value approximator*. We define the parameterized representation distance, $U_\omega$, as an approximant to $U^\pi$:

$$U^\pi(x,y) \approx U_\omega(x,y) := \frac{\|\phi_\omega(x)\|_2^2 + \|\phi_\omega(y)\|_2^2}{2} + \beta\theta(\phi_\omega(x),\phi_\omega(y))$$

where $\theta(\phi_\omega(x),\phi_\omega(y))$ is the angle between vectors $\phi_\omega(x)$ and $\phi_\omega(y)$ and $\beta$ is a scalar (in our results we use $\beta = 0.1$ but present results with other values of $\beta$ in the appendix).

Based on Equation (2), our learning target is then $T_{\bar\omega}^U(r_x,x',r_y,y') = |r_x - r_y| + \gamma U_{\bar\omega}(x',y')$, where $\bar\omega$ is a separate copy of the network parameters that are synchronised with $\omega$ at infrequent intervals. This is a common practice that was introduced by Mnih et al. [2015] (and in fact, we use the same update schedule they propose). The loss for this learning target is

$$\mathcal{L}_{\text{MICo}}(\omega) = \mathbb{E}_{\langle x,r_x,x'\rangle,\langle y,r_y,y'\rangle}\left[\left(T_{\bar\omega}^U(r_x,x',r_y,y') - U_\omega(x,y)\right)^2\right]$$

where $\langle x,r_x,x'\rangle$ and $\langle y,r_y,y'\rangle$ are pairs of transitions sampled from the agent's replay buffer. We can combine $\mathcal{L}_{\text{MICo}}$ with the temporal-difference loss $\mathcal{L}_{\text{TD}}$ of any RL agent as $(1-\alpha)\mathcal{L}_{\text{TD}} + \alpha\mathcal{L}_{\text{MICo}}$, where $\alpha \in (0,1)$. Each sampled mini-batch is used for both MICo and TD losses. Figure 3 (left) illustrates the network architecture used for learning.

Although the loss $\mathcal{L}_{\text{MICo}}$ is designed to learn the MICo diffuse metric $U^\pi$, the values of the metric itself are parametrised through $U_\omega$ defined above, which is constituted by several distinct terms. This appears to leave a question as to how the representations $\phi_\omega(x)$ and $\phi_\omega(y)$, as Euclidean vectors, are related to one another when the MICo loss is minimised. Careful inspection of the form of $U_\omega(x,y)$ shows that the (scaled) angular distance between $\phi_\omega(x)$ and $\phi_\omega(y)$ can be recovered from $U_\omega$ by subtracting the learnt approximations to the self-distances $U^\pi(x,x)$ and $U^\pi(y,y)$ (see Figure 3, right). We therefore define the reduced MICo distance $\Pi U^\pi$, which encodes the distances enforced between the representation vectors $\phi_\omega(x)$ and $\phi_\omega(y)$, by:

$$\beta\theta(\phi_\omega(x),\phi_\omega(y)) \approx \Pi U^\pi(x,y) = U^\pi(x,y) - \frac{1}{2}U^\pi(x,x) - \frac{1}{2}U^\pi(y,y).$$

In the following section we investigate the following two questions: **(1)** How informative of $V^\pi$ is $\Pi U^\pi$?; and **(2)** How useful are the features encountered by $\Pi U^\pi$ for policy evaluation? We conduct these investigations on tabular environments where we can compute the metrics exactly, which helps clarify the behaviour of our loss when combined with deep networks in Section 6.

### 5.1 Value bound gaps

Although Proposition 4.8 states that we have $|V^\pi(x) - V^\pi(y)| \leq U^\pi(x,y)$, we do not, in general, have the same upper bound for $\Pi U^\pi(x,y)$ as demonstrated by the following result.

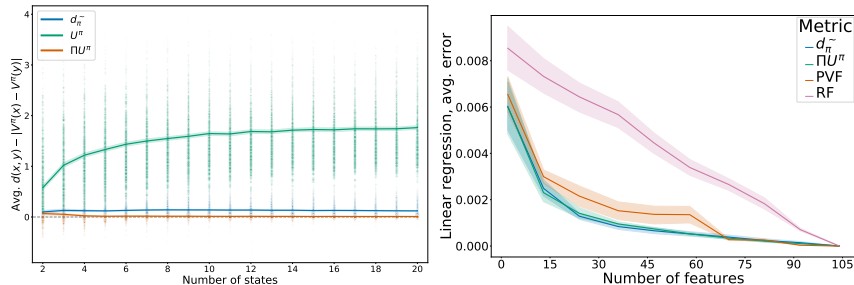

Figure 4: **Left:** The gap between the difference in values and the various distances for Garnet MDPs with varying numbers of actions (represented by circle sizes); **Right:** Average error when performing linear regression on varying numbers of features in the four-rooms GridWorld, averaged over 10 runs; shaded areas represent 95% confidence intervals.

**Lemma 5.1.** There exists an MDP with $x, y \in \mathcal{X}$, and $\pi \in \Pi$ where $|V^\pi(x) - V^\pi(y)| > \Pi U^\pi(x, y)$.

*Proof.* Consider a single-action MDP with two states ($x$ and $y$) where $y$ is absorbing, $x$ transitions with equal probability to $x$ and $y$, and a reward of $1$ is received only upon taking an action from state $x$. There is only one policy for this MDP which yields the value function $V(x) \approx 1.8$ and $V(y) = 0$. The MICo distance gives $U(x, x) \approx 1.06$, $U(x, y) \approx 1.82$, and $U(y, y) = 0$, while the reduced MICo distance yields $\Pi U(x, x) = \Pi U(y, y) = 0$, and $\Pi U(x, y) \approx 1.29 < |V(x) - V(y)| = 1.8$. $\square$

Despite this negative result, it is worth evaluating how often *in practice* this inequality is violated and by how much, as this directly impacts the utility of this distance for learning representations.

To do so, we make use of *Garnet MDPs*, a class of randomly generated MDPs [Archibald et al., 1995, Piot et al., 2014]. Given a specified number of states $n_\mathcal{X}$ and the number of actions $n_\mathcal{A}$, Garnet($n_\mathcal{X}, n_\mathcal{A}$) is generated as follows: **1.** The branching factor $b_{x,a}$ of each transition $P_x^a$ is sampled uniformly from $[1 : n_\mathcal{X}]$. **2.** $b_{x,a}$ states are picked uniformly randomly from $\mathcal{X}$ and assigned a random value in $[0, 1]$; these values are then normalized to produce a proper distribution $P_x^a$. **3.** Each $r_x^a$ is sampled uniformly in $[0, 1]$.

For each Garnet($n_\mathcal{X}, n_\mathcal{A}$) we sample 100 stochastic policies $\{\pi_i\}$ and compute the average gap: $\frac{1}{100|\mathcal{X}|^2} \sum_i \sum_{x,y} d(x, y) - |V^{\pi_i}(x) - V^{\pi_i}(y)|$, where $d$ stands for any of the considered metrics. Note we are measuring the *signed* difference, as we are interested in the frequency with which the upper bound is violated. As seen in Figure 4 (left), our metric *does* on average provide an upper bound on the difference in values that is also tighter bound than those provided by $U^\pi$ and $\pi$-bisimulation. This suggests that the resulting representations remain informative of value similarities.

## 5.2 State features

In order to investigate the usefulness of the representations produced by $\Pi U^\pi$, we construct state features directly by using the computed distances to project the states into a lower-dimensional space with the UMAP dimensionality reduction algorithm [McInnes et al., 2018][2]. We then apply linear regression of the true value function $V^\pi$ against the features to compute $\hat{V}^\pi$ and measure the average error across the state space. As baselines we compare against random features (RF), Proto Value Functions (PVF) [Mahadevan and Maggioni, 2007], and the features produced by $\pi$-bisimulation [Castro, 2020]. We present our results on the well-known four-rooms GridWorld [Sutton et al., 1999] in Figure 4 (right) and provide results on more environments in the appendix. Despite the independent couplings, $\Pi U^\pi$ performs on par with $\pi$-bisimulation, which optimizes over all couplings.

## 6 Large-scale empirical evaluation

Having developed a greater understanding of the properties inherent to the representations produced by the MICo loss, we evaluate it on the Arcade Learning Environment [Bellemare et al., 2013]. We added the MICo loss to all the JAX agents provided in the Dopamine library [Castro et al., 2018]: DQN [Mnih et al., 2015], Rainbow [Hessel et al., 2018], QR-DQN [Dabney et al., 2018b],

---

[2]Note that since UMAP expects a metric, it is ill-defined with the diffuse metric $U^\pi$.

and IQN [Dabney et al., 2018a], using mean squared error loss to minimize $\mathcal{L}_{\text{TD}}$ for DQN (as suggested by Obando-Ceron and Castro [2021]). Given the state-of-the-art results demonstrated by the Munchausen-IQN (M-IQN) agent [Vieillard et al., 2020], we also evaluated incorporating our loss into M-IQN.[3] For all experiments we used the hyperparameter settings provided with Dopamine. We found that a value of $\alpha = 0.5$ worked well with quantile-based agents (QR-DQN, IQN, and M-IQN), while a value of $\alpha = 0.01$ worked well with DQN and Rainbow. We hypothesise that the difference in scale of the quantile, categorical, and non-distributional loss functions concerned leads to these distinct values of $\alpha$ performing well. We found it important to use the Huber loss [Huber, 1964] to minimize $\mathcal{L}_{\text{MICo}}$ as this emphasizes greater accuracy for smaller distances as opposed to larger distances. We experimented using the MSE loss but found that larger distances tended to overwhelm the optimization process, thereby degrading performance.

We evaluated on all 60 Atari 2600 games over 5 seeds and report the results in Figure 1 (left), using the interquantile metric (IQM), proposed by Agarwal et al. [2021b] as a more robust and reliable alternative to mean and median (which are reported in Figure 6). The fact that the MICo loss provides consistent improvements over a wide range of baseline agents of varying complexity suggests that the MICo loss can help learn better representations for control.

Additionally, we evaluated the MICo loss on twelve of the DM-Control suite from pixels environments [Tassa et al., 2018]. As a base agent we used Soft Actor-Critic (SAC) [Haarnoja et al., 2018] with the convolutional auto-encoder described by Yarats et al. [2019]. We applied the MICo loss on the output of the auto-encoder (with $\alpha = 1e-5$) and maintained all other parameters untouched. Recently, Zhang et al. [2021] introduced DBC, which learns a dynamics and reward model on the output of the auto-encoder; their bisimulation loss uses the learned dynamics model in the computation of the Kantorovich distance between the next state transitions. We consider two variants of their algorithm: one which learns a stochastic dynamics model (DBC), and one which learns a deterministic dynamics model (DBC-Det). We replaced their bisimulation loss with the MICo loss (which, importantly, does not require a dynamics model) and kept all other parameters untouched. As Figure 1 illustrates, the best performance is achieved with SAC augmented with the MICo loss; additionally, replacing the bisimulation loss of DBC with the MICo loss is able to recover the performance of DBC to match that of SAC.

Additional details and results are provided in the appendix.

## 7   Related Work

Bisimulation metrics were introduced for MDPs by Ferns et al. [2004], and have been extended in a number of directions [Ferns et al., 2005, 2006, Taylor, 2008, Taylor et al., 2009, Ferns et al., 2011, Comanici et al., 2012, Bacci et al., 2013a,b, Abate, 2013, Ferns and Precup, 2014, Castro, 2020], with applications including policy transfer [Castro and Precup, 2010, Santara et al., 2019], representation learning [Ruan et al., 2015, Comanici et al., 2015], and state aggregation [Li et al., 2006].

A range of other notions of similarity in MDPs have also been considered, such as action sequence equivalence [Givan et al., 2003], temporally extended metrics [Amortila et al., 2019], MDP homomorphisms [Ravindran and Barto, 2003], utile distinction [McCallum, 1996], and policy irrelevance [Jong and Stone, 2005], as well as notions of policy similarity [Pacchiano et al., 2020, Moskovitz et al., 2021]. Li et al. [2006] review different notions of similarity applied to state aggregation. Recently, Le Lan et al. [2021] performed an exhaustive analysis of the continuity properties, relative to functions of interest in RL, of a number of existing metrics in the literature.

The notion of zero self-distance, central to the diffuse metrics defined in this paper, is increasingly encountered in machine learning applications involving approximation of losses. Of particular note is entropy-regularised optimal transport [Cuturi, 2013] and related quantities [Genevay et al., 2018, Fatras et al., 2020, Chizat et al., 2020, Fatras et al., 2021].

More broadly, many approaches to representation learning in deep RL have been considered, such as those based on auxiliary tasks (see e.g. [Sutton et al., 2011, Jaderberg et al., 2017, Bellemare et al.,

---

[3]Given that the authors of M-IQN had implemented their agent in TensorFlow (whereas our agents are in JAX), we have reimplemented M-IQN in JAX and run 5 independent runs (in contrast to the 3 run by Vieillard et al. [2020].

2017, François-Lavet et al., 2019, Gelada et al., 2019, Guo et al., 2020b]), and other approaches such as successor features [Dayan, 1993, Barreto et al., 2017].

# 8 Conclusion

In this paper, we have introduced the MICo distance, a notion of state similarity that can be learnt at scale and from samples. We have studied the theoretical properties of MICo, and proposed a new loss to make the non-zero self-distances of this diffuse metric compatible with function approximation, combining it with a variety of deep RL agents to obtain strong performance on the Arcade Learning Environment. In contrast to auxiliary losses that *implicitly* shape an agent's representation, MICo directly modifies the features learnt by a deep RL agent; our results indicate that this helps improve performance. To the best of our knowledge, this is the first time *directly* shaping the representation of RL agents has been successfully applied at scale. We believe this represents an interesting new approach to representation learning in RL; continuing to develop theory, algorithms and implementations for direct representation shaping in deep RL is an important and promising direction for future work.

**Broader impact statement**

This work lies in the realm of "foundational RL" in that it contributes to the fundamental understanding and development of reinforcement learning algorithms and theory. As such, despite us agreeing in the importance of this discussion, our work is quite far removed from ethical issues and potential societal consequences.

# 9 Acknowledgements

The authors would like to thank Gheorghe Comanici, Rishabh Agarwal, Nino Vieillard, and Matthieu Geist for their valuable feedback on the paper and experiments. Pablo Samuel Castro would like to thank Roman Novak and Jascha Sohl-Dickstein for their help in getting angular distances to work stably! Thanks to Hongyu Zang for pointing out that the x-axis labels for the SAC experiments needed to be fixed. Finally, the authors would like to thank the reviewers (both ICML'21 and NeurIPS'21) for helping make this paper better.

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
