# Supplementary Material:
# MICo: Improved representations via sampling-based state similarity for Markov decision processes

## A  Extended background material

In this section we provide a more extensive background review.

### A.1  Reinforcement learning

In this section we give a slightly more expansive overview of relevant key concepts in reinforcement learning, without the space constraints of the main paper. Denoting by $\mathscr{P}(S)$ the set of probability distributions on a set $S$, we define a Markov decision process $(\mathcal{X}, \mathcal{A}, \gamma, P, r)$ as:

- A finite state space $\mathcal{X}$;
- A finite action space $\mathcal{A}$;
- A transition kernel $P : \mathcal{X} \times \mathcal{A} \to \mathscr{P}(\mathcal{X})$;
- A reward function $r : \mathcal{X} \times \mathcal{A} \to \mathbb{R}$;
- A discount factor $\gamma \in [0, 1)$.

For notational convenience we introduce the notation $P_x^a \in \mathscr{P}(\mathcal{X})$ for the next-state distribution given state-action pair $(x, a)$, and $r_x^a$ for the corresponding immediate reward.

Policies are mappings from states to distributions over actions: $\pi \in \mathscr{P}(\mathcal{A})^{\mathcal{X}}$ and induce a *value function* $V^\pi : \mathcal{X} \to \mathbb{R}$ defined via the recurrence:

$$V^\pi(x) := \mathbb{E}_{a \sim \pi(x)} \left[ r_x^a + \gamma \mathbb{E}_{x' \sim P_x^a} [V^\pi(x')] \right] .$$

It can be shown that this recurrence uniquely defines $V^\pi$ through a contraction mapping argument [Bertsekas and Tsitsiklis, 1996].

The control problem is concerned with finding the optimal policy

$$\pi^* = \arg \max_{\pi \in \mathscr{P}(\mathcal{A})^{\mathcal{X}}} V^\pi .$$

It can be shown that while the optimisation problem above appears to have multiple objectives (one for each coordinate of $V^\pi$, there is in fact a policy $\pi^* \in \mathscr{P}(\mathcal{A})^{\mathcal{X}}$ that simultaneously maximises all coordinates of $V^\pi$, and that this policy can be taken to be deterministic; that is, for each $x \in \mathcal{X}$, $\pi(\cdot|x) \in \mathscr{P}(\mathcal{A})$ attributes probability 1 to a single action. In reinforcement learning in particular, we are often interested in finding, or approximating, $\pi^*$ from direct interaction with the MDP in question via sample trajectories, *without knowledge of $P$ or $r$* (and sometimes not even $\mathcal{X}$).

### A.2  Metrics

A *metric* $d$ on a set $X$ is a function $d : X \times X \to [0, \infty)$ respecting the following axioms for any $x, y, z \in X$:

1. **Identity of indiscernibles:** $d(x, y) = 0 \iff x = y$;
2. **Symmetry:** $d(x, y) = d(y, x)$;
3. **Triangle inequality:** $d(x, y) \le d(x, z) + d(z, y)$.

A *pseudometric* is similar, but the "identity of indiscernibles" axiom is weakened:

1. $x = y \implies d(x, y) = 0$;
2. $d(x, y) = d(y, x)$;
3. $d(x, y) \le d(x, z) + d(z, y)$.

Note that the weakened first condition *does* allow one to have $d(x, y) = 0$ when $x \neq y$.

A *(pseudo)metric space* $(X, d)$ is defined as a set $X$ together with a (pseudo)metric $d$ defined on $X$.

## A.3 State similarity and bisimulation metrics

Bisimulation is a fundamental notion of behavioural equivalence introduced by Park and Milner [Milner, 1989] in the early 1980s in the context of nondeterministic transition systems. The probabilistic analogue was introduced by Larsen and Skou [1991]. The notion of an equivalence relation is not suitable to capture the extent to which quantitative systems may resemble each other in behaviour. To provide a quantitative notion, bisimulation metrics were introduced by Desharnais et al. [1999, 2004] in the context of probabilistic transition systems without rewards. In reinforcement learning the reward is an important ingredient, accordingly the *bisimulation metric* for states of MDPs was introduced by Ferns et al. [2004]. Much work followed this initial introduction of bisimulation metrics into RL, as described in the main paper. We briefly reviewed the bisimulation metric in Section 2, and now provide additional detail around some of the key associated mathematical concepts.

Central to the definition of the bisimulation metric is the operator $T_k : \mathcal{M}(\mathcal{X}) \to \mathcal{M}(\mathcal{X})$, defined over $\mathcal{M}(\mathcal{X})$, the space of pseudometrics on $\mathcal{X}$. Pseudometrics were explored in more detail in Section A.2. We now turn to the definition of the operator itself, given by

$$T_k(d)(x,y) = \max_{a \in \mathcal{A}}[|r_x^a - r_y^a| + \gamma W_d(P_x^a, P_y^a)],$$

for each $d \in \mathcal{M}(\mathcal{X})$, and each $x, y \in \mathcal{X}$. It can be verified that the function $T_K(d) : \mathcal{X} \times \mathcal{X} \to \mathbb{R}$ satisfies the properties of a pseudometric, so under this definition $T_K$ does indeed map $\mathcal{M}(\mathcal{X})$ into itself.

The other central mathematical concept underpinning the operator $T_K$ is the Wasserstein distance $W_d$ using base metric $d$. $W_d$ is formally a pseudometric over the set of probability distributions $\mathscr{P}(\mathcal{X})$, defined as the solution to an optimisation problem. The problem specifically is formulated as finding an optimal coupling between the two input probability distributions that minimises a notion of transport cost associated with $d$. Mathematically, for two probability distributions $\mu, \mu' \in \mathscr{P}(\mathcal{X})$, we have

$$W_d(\mu, \mu') = \min_{\substack{(Z,Z') \\ Z \sim \mu, Z' \sim \nu'}} \mathbb{E}[d(Z, Z')].$$

Note that the pair of random variables $(Z, Z')$ attaining the minimum in the above expression will in general not be independent. That the minimum is actually attained in the above example in the case of a finite set $\mathcal{X}$ can be seen by expressing the optimisation problem as a linear program. Minima are obtained in much more general settings too; see Villani [2008].

Finally, the operator $T_K$ can be analysed in a similar way to standard operators in dynamic programming for reinforcement learning. It can be shown that it is a contraction mapping with respect to the $L^\infty$ metric over $\mathcal{M}(\mathcal{X})$, and that $\mathcal{M}(\mathcal{X})$ is a complete metric space with respect to the same metric [Ferns et al., 2011]. Thus, by Banach's fixed point theorem, $T_K$ has a unique fixed point in $\mathcal{M}(\mathcal{X})$, and repeated application of $T_K$ to any initial pseudometric will converge to this fixed point.

## A.4 Further details on diffuse and partial metrics

The notion of a distance function having non-zero self distance was first introduced by Matthews [1994] who called it a *partial metric*. We define it below:

**Definition A.1.** Given a set $\mathcal{X}$, a function $d : \mathcal{X} \times \mathcal{X} \to \mathbb{R}$ is a partial metric if the following axioms hold: (i) $x = y \iff d(x,x) = d(y,y) = d(x,y)$ for any $x, y \in \mathcal{X}$; (ii) $d(x,x) \le d(y,x)$ for any $x, y \in \mathcal{X}$; (iii) $d(x,y) = d(y,x)$ for any $x, y \in \mathcal{X}$; (iv) $d(x,y) \le d(x,z) + d(y,z) - d(z,z)$ $\forall x, y, z \in \mathcal{X}$.

This definition was introduced to recover a proper metric from the distance function: that is, given a partial metric $d$, one is guaranteed that $\tilde{d}(x,y) = d(x,y) - \frac{1}{2}(d(x,x) + d(y,y))$ is a proper metric.

The above definition is still too stringent for the Łukaszyk–Karmowski distance (and hence MICo distance), since it fails axiom 4 as shown in the following counterexample.

**Example A.2.** The Łukaszyk–Karmowski distance does not satisfy the modified triangle inequality: let $\mathcal{X}$ be $[0, 1]$, and $\rho$ be the Euclidean distance $|\cdot|$. Let $\mu, \nu$ be Dirac measures concentrated at 0 and 1, and let $\eta$ be $\frac{1}{2}(\delta_0 + \delta_1)$. Then one can calculate that $d_{LK}(\rho)(\mu, \nu) = 1$, while $d_{LK}(\rho)(\mu, \eta) + d_{LK}(\rho)(\nu, \eta) - d_{LK}(\rho)(\eta, \eta) = 1/2$, breaking the inequality.

This naturally leads us to the notion of diffuse metrics defined in the main paper.

# B    Proof of Proposition 4.2

**Proposition 4.2.** The operator $T_M^\pi$ is a contraction mapping on $\mathbb{R}^{\mathcal{X} \times \mathcal{X}}$ with respect to the $L^\infty$ norm.

*Proof.* Let $U, U' \in \mathbb{R}^{\mathcal{X} \times \mathcal{X}}$. Then note that

$$|(T^\pi U)(x,y) - (T^\pi U')(x,y)| = \left| \gamma \sum_{x', y'} \pi(a|x) \pi(b|y) P_x^a(x') P_y^b(y') (U - U')(x', y') \right| \le \gamma \|U - U'\|_\infty .$$

for any $x, y \in \mathcal{X}$, as required.                                                   $\square$

# C    Experimental details

We will first describe the regular network and training setup for these agents so as to facilitate the description of our loss.

## C.1    Baseline network and loss description

The networks used by Dopamine for the ALE consist of 3 convolutional layers followed by two fully-connected layers (the output of the networks depends on the agent). We denote the output of the convolutional layers by $\phi_\omega$ with parameters $\omega$, and the remaining fully connected layers by $\psi_\xi$ with parameters $\xi$. Thus, given an input state $x$ (e.g. a stack of 4 Atari frames), the output of the network is $Q_{\xi,\omega}(x, \cdot) = \psi_\xi(\phi_\omega(x))$. Two copies of this network are maintained: an *online* network and a *target* network; we will denote the parameters of the target network by $\bar{\xi}$ and $\bar{\omega}$. During learning, the parameters of the online network are updated every 4 environment steps, while the target network parameters are synced with the online network parameters every 8000 environment steps. We refer to the loss used by the various agents considered as $\mathcal{L}_{\text{TD}}$; for example, for DQN this would be:

$$\mathcal{L}_{\text{TD}}(\xi, \omega) := \mathbb{E}_{(x,a,r,x') \sim \mathcal{D}} \left[ \rho \left( r + \gamma \max_{a' \in \mathcal{A}} Q_{\bar{\xi}, \bar{\omega}}(x', a') - Q_{\xi, \omega}(x, a) \right) \right] ,$$

where $\mathcal{D}$ is a replay buffer with a capacity of 1M transitions, and $\rho$ is the Huber loss.

## C.2    MICo loss description

We will be applying the MICo loss to $\phi_\omega(x)$. As described in Section 5, we express the distance between two states as:

$$U_\omega(x, y) = \frac{\|\phi_\omega(x)\|_2^2 + \|\phi_{\bar{\omega}}(y)\|_2^2}{2} + \beta \theta(\phi_\omega(x), \phi_{\bar{\omega}}(y)) ,$$

where $\theta(\phi_\omega(x), \phi_{\bar{\omega}}(y))$ is the angle between vectors $\phi_\omega(x)$ and $\phi_{\bar{\omega}}(y)$ and $\beta$ is a scalar. Note that we are using the target network for the $y$ representations; this was done for learning stability. We used $\beta = 0.1$ for the results in the main paper, but present some results with different values of $\beta$ below.

In order to get a numerically stable operation, we implement the angular distance between representations $\phi_\omega(x)$ and $\phi_\omega(y)$ according to the calculations

$$\text{CS}(\phi_\omega(x), \phi_\omega(y)) = \frac{\langle \phi_\omega(x), \phi_\omega(y) \rangle}{\|\phi_\omega(x)\| \|\phi_\omega(y)\|}$$

$$\theta(\phi_\omega(x), \phi_\omega(y)) = \text{arctan2} \left( \sqrt{1 - \text{CS}(\phi_\omega(x), \phi_\omega(y))^2}, \text{CS}(\phi_\omega(x), \phi_\omega(y)) \right) .$$

Based on Equation (2), our learning target is then (note the target network is used for both representations here):

$$T_{\bar{\omega}}^U(r_x, x', r_y, y') = |r_x - r_y| + \gamma U_{\bar{\omega}}(x', y') ,$$

and the loss is

$$\mathcal{L}_{\text{MICo}}(\omega) = \mathbb{E}_{\substack{\langle x, r_x, x' \rangle \sim \mathcal{D} \\ \langle y, r_y, y' \rangle}} \left[ \left( T_{\bar{\omega}}^U (r_x, x', r_y, y') - U_\omega(x, y) \right)^2 \right] ,$$

As mentioned in Section 5, we use the same mini-batch sampled for $\mathcal{L}_{\text{TD}}$ for computing $\mathcal{L}_{\text{MICo}}$. Specifically, we follow the method introduced by Castro [2020] for constructing new matrices that allow us to compute the distances between all pairs of sampled states (see code for details on matrix operations). Our combined loss is then $\mathcal{L}_\alpha(\xi, \omega) = (1 - \alpha)\mathcal{L}_{\text{TD}}(\xi, \omega) + \alpha \mathcal{L}_{\text{MICo}}(\omega)$.

### C.3  Hyperparameters for soft actor-critic

We re-implemented the DBC algorithm from Zhang et al. [2021] on top of the Soft Actor-Critic algorithm [Haarnoja et al., 2018] provided by the Dopamine library [Castro et al., 2018]. We compared the following algorithms, using the same hyperparameters for all[4]:

1. **SAC:** This is Soft Actor-Critic [Haarnoja et al., 2018] with the convolutional encoder described by Yarats et al. [2019].

2. **DBC:** This is DBC, based on SAC, as described by Zhang et al. [2021].

3. **DBC-Det:** In the code provided by Zhang et al. [2021], the default setting was to assume deterministic transitions (which is an easier dynamics model to learn), so we decided to compare against this version as well. It is interesting to note that the performance is roughly the same as for DBC.

4. **MICo:** This a modified version of SAC, adding the MICo loss to the output of the encoder. Note that the encoder output is the same one used by DBC for their dynamics and reward models.

5. **DBC+MICo:** Instead of using the bisimulation loss of Zhang et al. [2021], which relies on the learned dynamics model, we use our MICo loss. We kept all other components untouched (so a dynamics and reward model were still being learned).

It is worth noting that some of the hyperparameters we used differ from those listed in the code provided by Zhang et al. [2021]; in our experiments they hyperparameters for all agents are based on the default SAC hyperparameters in the Dopamine library [Castro et al., 2018].

For the ALE experiments we used the "squaring" of the sampled batches introduced by Castro [2020] (where all pairs of sampled states are considered). However, the implementation provided by Zhang et al. [2021] instead created a copy of the sampled batch of transitions and shuffled them; we chose to follow this setup for the SAC-based experiments. Thus, while in the ALE experiments we are comparing $m^2$ pairs of states (where $m$ is the batch size) at each training step, in the SAC-based experiments we are only comparing $m$ pairs of states.

The aggregate results are displayed in Figure 1, and per-environment results in Figure 14 below.

## D  Additional experimental results

### D.1  Additional state feature results

The results shown in Figure 4 are on the well-known four-rooms GridWorld [Sutton et al., 1999]. We provide extra experiments in Figure 5.

### D.2  Complete ALE experiments

We additionally provide complete results for all the agents in Figure 8, Figure 9, Figure 10, Figure 11, and Figure 12.

---

[4]See https://github.com/google-research/google-research/tree/master/mico for all hyperparameter settings.

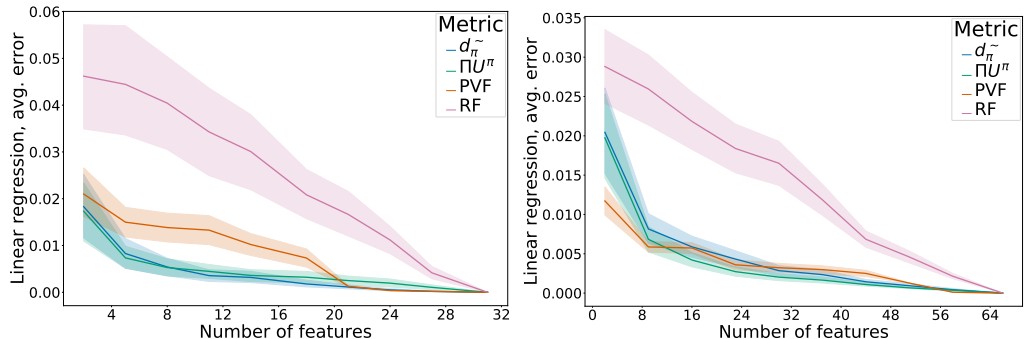

Figure 5: Average error when performing linear regression on varying numbers of features on the mirrored rooms introduced by Castro [2020] (left) and the grid task introduced by Dayan [1993] (right). Averaged over 10 runs; shaded areas represent 95% confidence intervals.

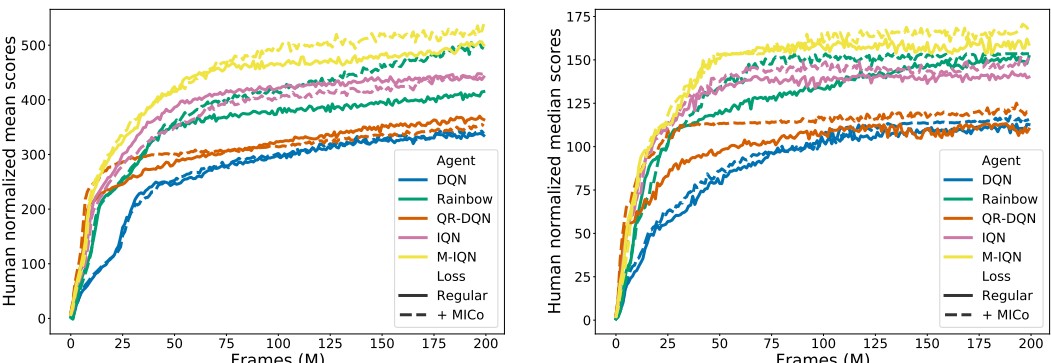

Figure 6: Mean (left) and median (right) human normalized scores across 60 Atari 2600 games, averaged over 5 independent runs.

### D.3   Sweep over $\alpha$ and $\beta$ values

In Figure 13 we demonstrate the performance of the MICo loss when added to Rainbow over a number of different values of $\alpha$ and $\beta$. For each agent, we ran a similar hyperparameter sweep over $\alpha$ and $\beta$ on the same six games displayed in Figure 13 to determine settings to be used in the full ALE experiments.

### D.4   Complete DM-Control results

Full per-environment results are provided in Figure 14.

### D.5   Compute time and infrastructure

For Figure 4 each run took approximately 10 minutes. For Figure 4 and Figure 5 the running time varied for each environment and per metric but a conservative estimate is 30 minutes per run. All GPU experiments were run on NVIDIA Tesla P100 GPUs. Each Atari game takes approximately 5 days (300 hours) to run for 200M frames.

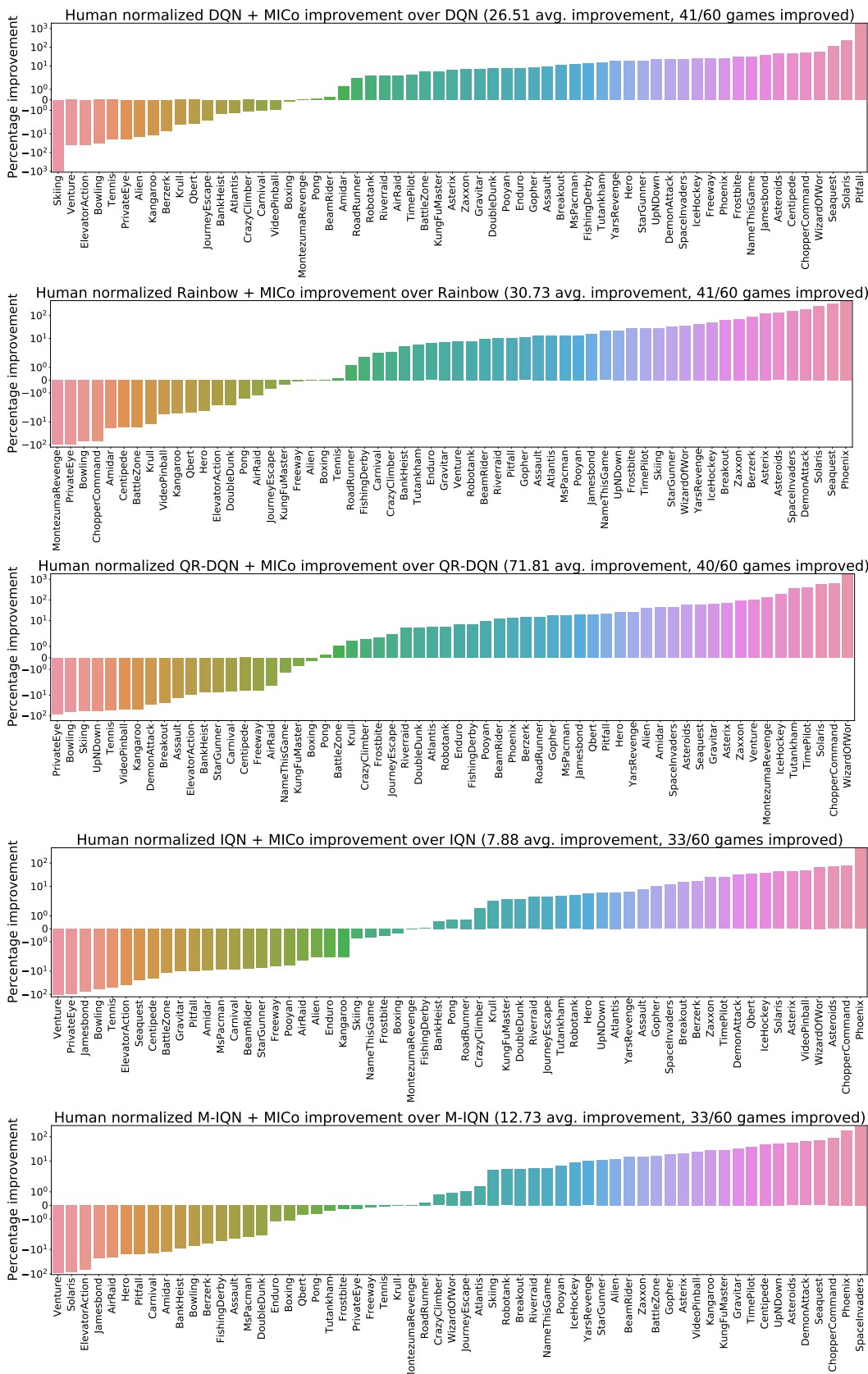

Figure 7: From top to bottom, percentage improvement in returns (averaged over the last 5 iterations) when adding $\mathcal{L}_{\text{MICo}}$ to DQN, Rainbow, QR-DQN, IQN, and M-DQN. The results for are averaged over 5 independent runs.

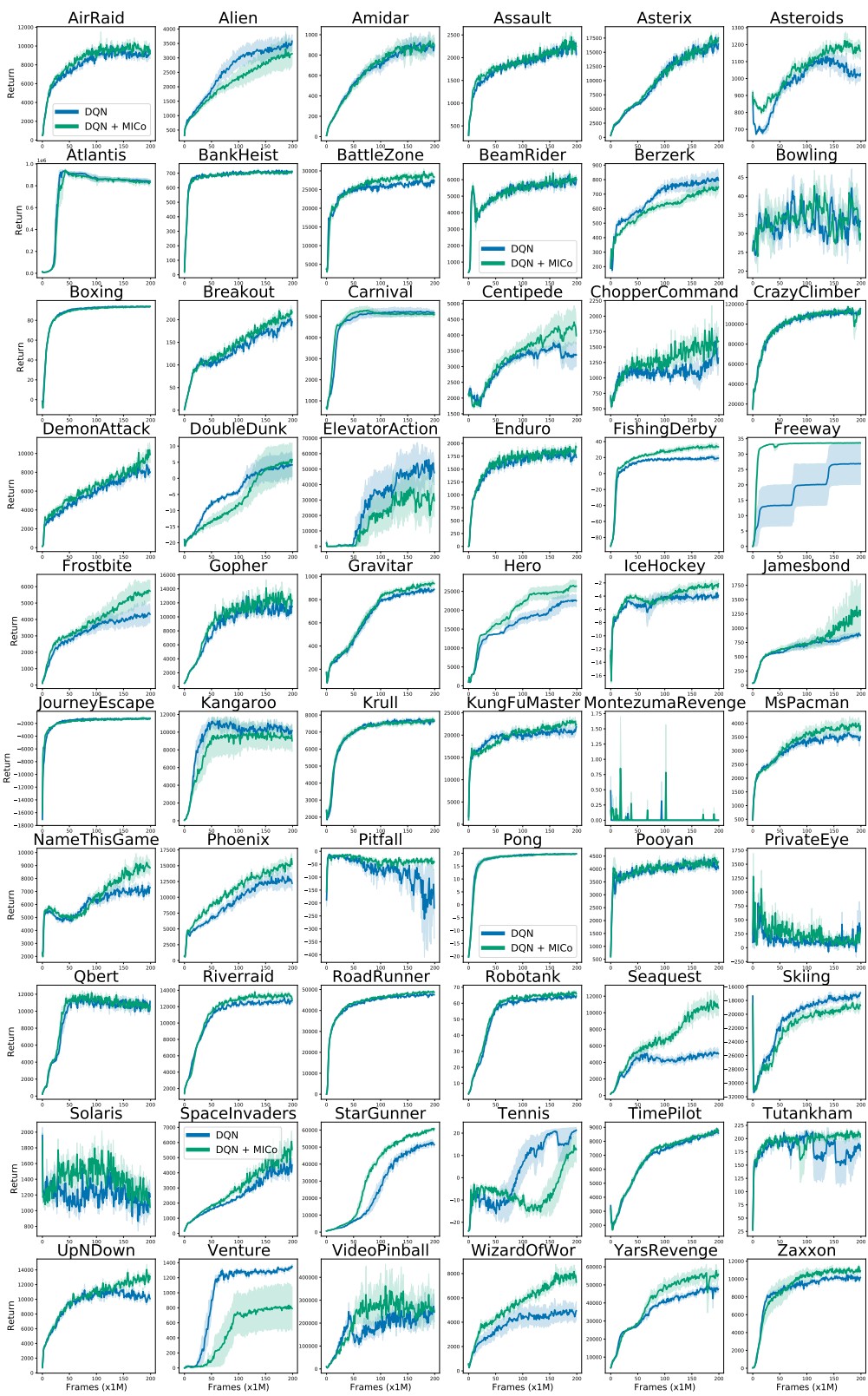

Figure 8: Training curves for DQN agents. The results for all games and agents are over 5 independent runs, and shaded regions report 75% confidence intervals.

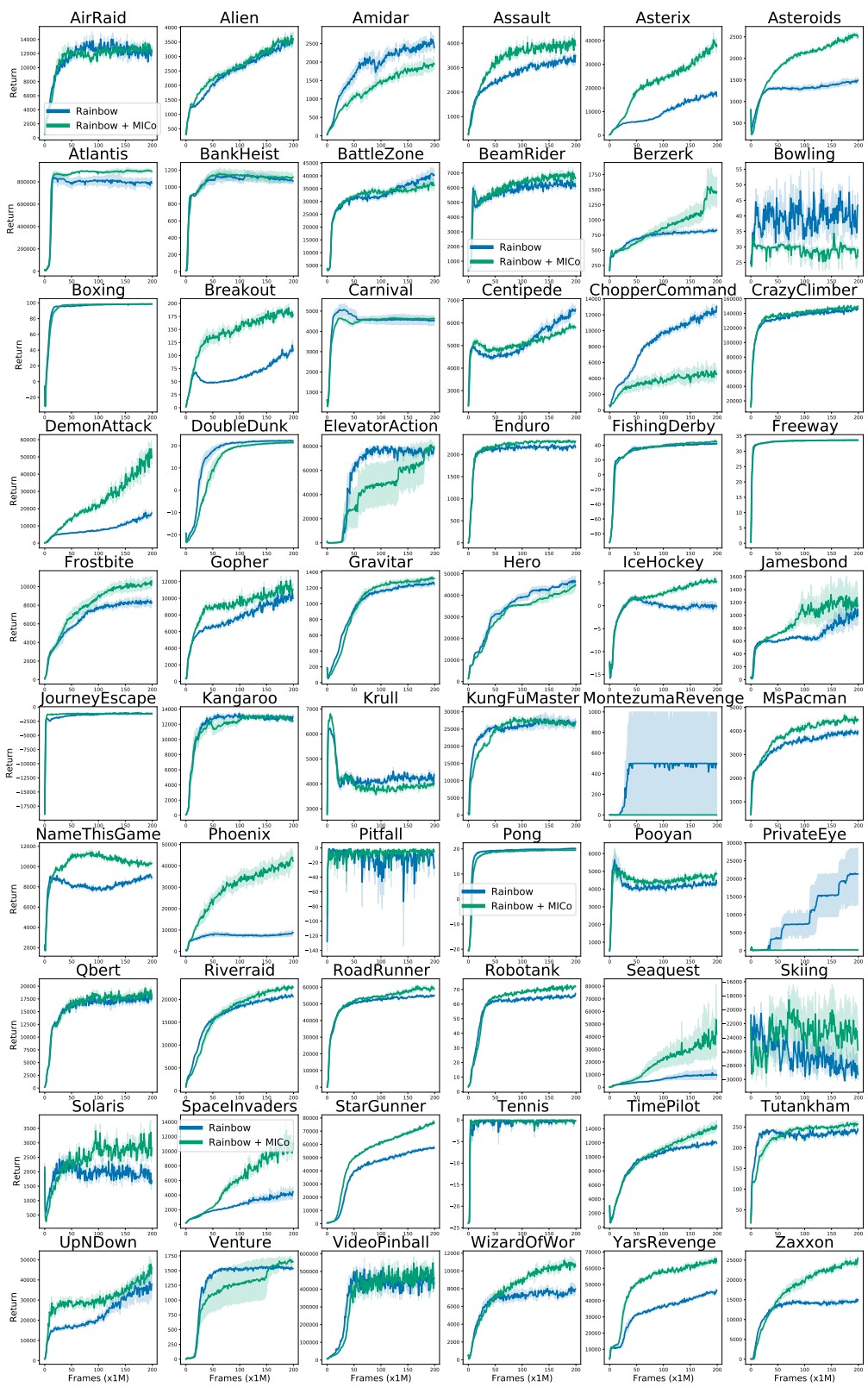

Figure 9: Training curves for Rainbow agents. The results for all games and agents are over 5 independent runs, and shaded regions report 75% confidence intervals.

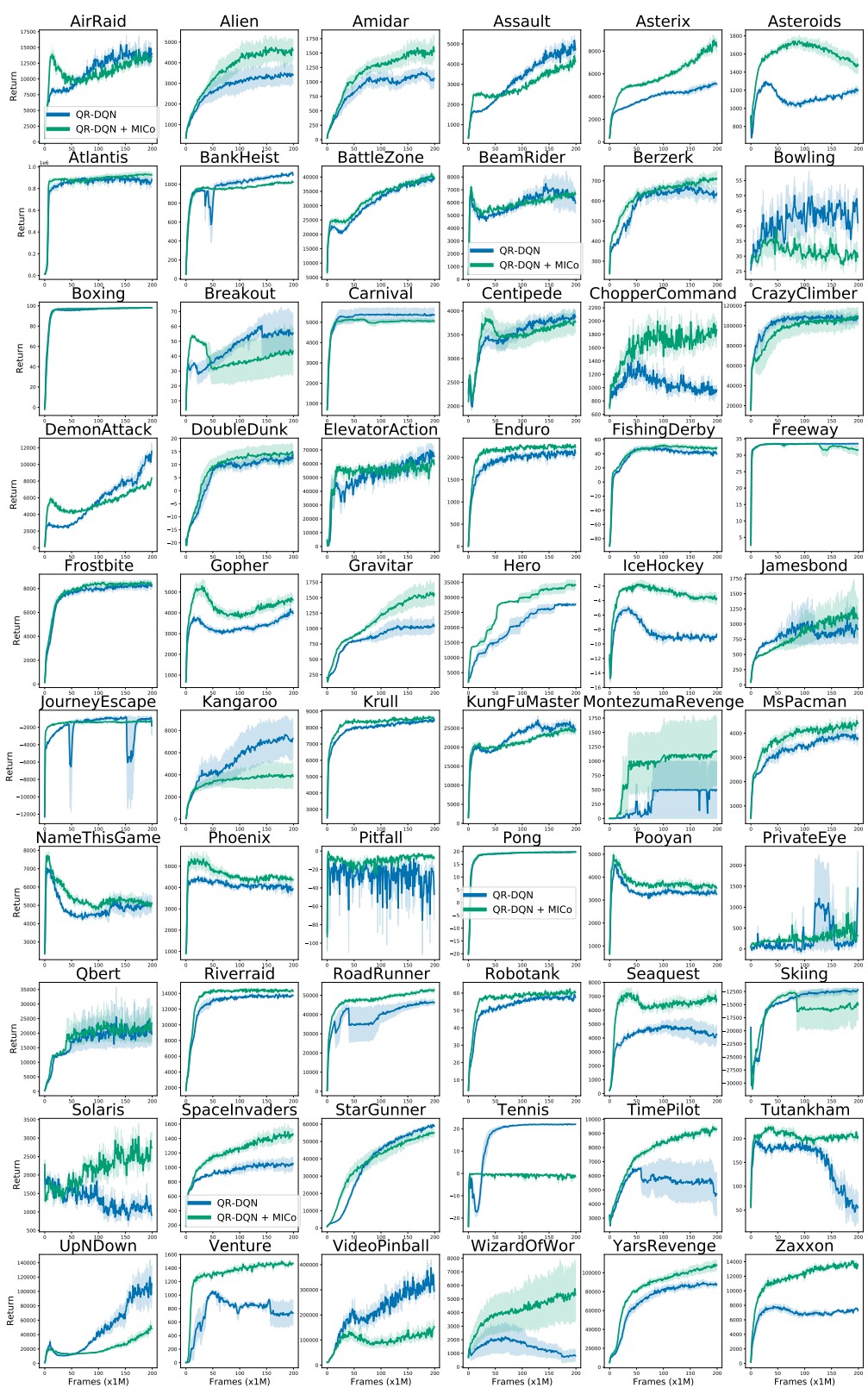

Figure 10: Training curves for QR-DQN agents. The results for all games and agents are over 5 independent runs, and shaded regions report 75% confidence intervals.

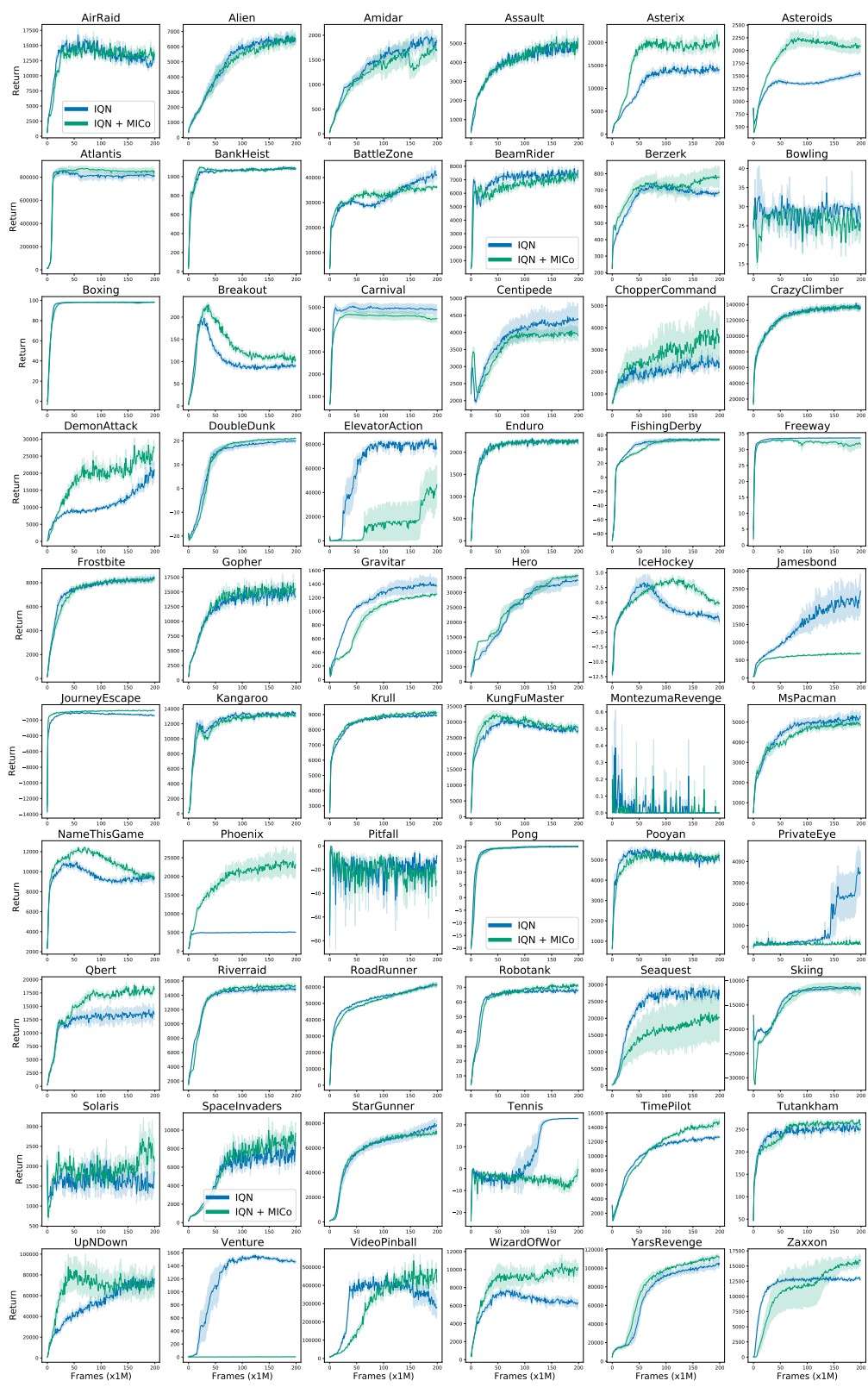

Figure 11: Training curves for IQN agents. The results for all games and agents are over 5 independent runs, and shaded regions report 75% confidence intervals.

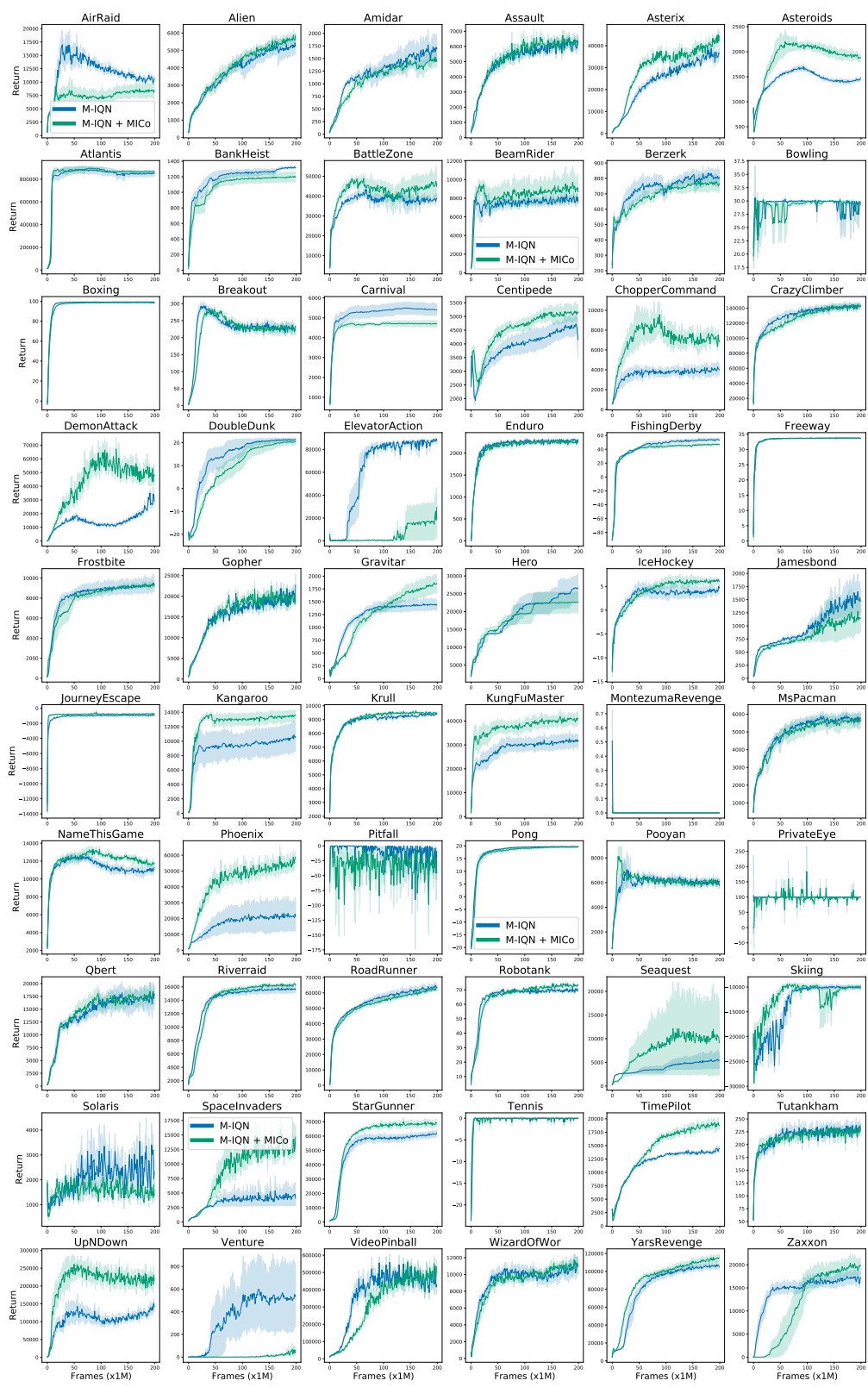

Figure 12: Training curves for M-IQN agents. The results for all games and agents are over 5 independent runs, and shaded regions report 75% confidence intervals.

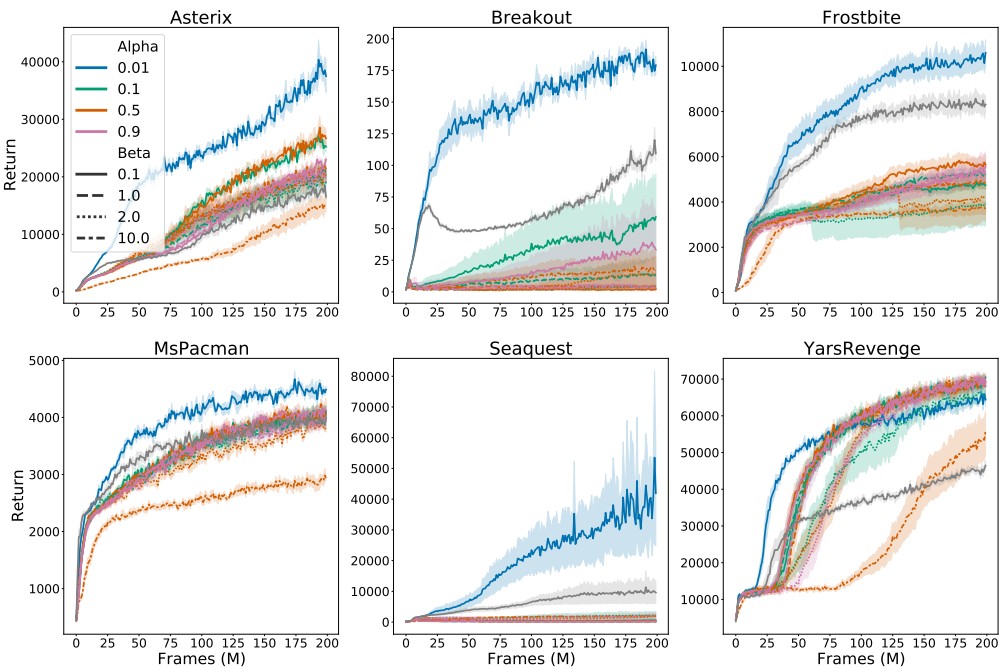

Figure 13: Sweeping over various values of $\alpha$ and $\beta$ when adding the MICo loss to Rainbow. The grey line represents regular Rainbow.

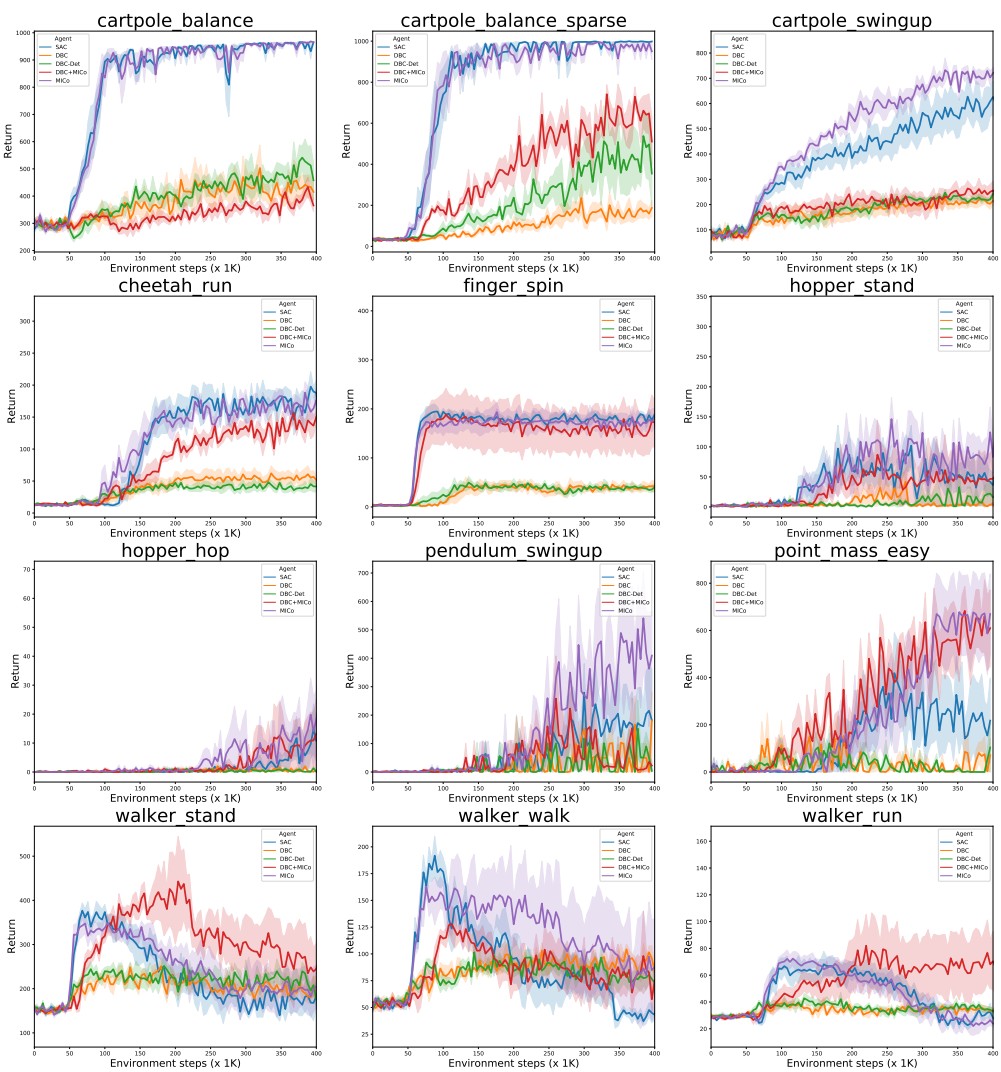

Figure 14: Comparison of all agents on twelve of the DM-Control suite. Each algorithm and environment was run for five independent seeds, and the shaded areas report 75% confidence intervals.