# OpenReview forum: "MICo: Improved representations via sampling-based state similarity for Markov decision processes"
_NeurIPS.cc/2021/Conference — NeurIPS 2021 Poster_

### Official Review · Reviewer_dAxX · 2021-07-04

**Rating:** 7
**Confidence:** 3

**Summary:**

This paper introduces MICo, a new behavioural distance over the state space of an MDP. Compared to previous works on behavioural metrics such as bisimulation metrics, MICo is compatible with sample-based algorithms and therefore can be easily integrated with deep RL methods. Empirical results demonstrate that learning this distance improves performance on the Atari benchmark.

**Limitations And Societal Impact:**

Yes

**Main Review:**

Overall, I think the paper is of high quality. The proposed method is well motivated and theoretically sound. The structure and writing are very clear. Nevertheless, I have some questions and hope the authors can help address.

* The flow of the paper is to first identify the drawbacks of the bisimulation metrics and then propose MICo to resolve them. However, it seems to me that it is more like proposing a new behavioural distance that "bypasses" the drawbacks of the bisimulation metrics. "Addressing the drawbacks" sounds like MICo can achieve the exactly same results as bisimulation metrics without suffering the drawbacks. I guess this is not what the authors want to convey. I would like to see some more discussion on comparing MICo and bisimulation metrics. For example, with the advantages of MICo being said, what does it lose (or fail to capture) compared to bisimulation metrics? In addition to Section 5.2, it would be better to add some visualizations of the MICo and bisimulation metrics (on a small grid world env), so that readers may get a more intuitive understanding.
* I do not quite agree with the description in Line 85 that explicitness means not using additional network parameters. Whether or not using additional network parameters is more on the implementation side.
* The fixed point $U^\pi$ is defined w.r.t. a policy $\pi$. However, when learning together with TD loss (as in Atari experiments), the policy keep changing. I am wondering how to deal with such non-stationarity.

Possible typos (please correct me if I am wrong):

* Line 163: "functional $T_K$ of $T_K$"
* Line 59: "$d$ symmetric", missing "is"?

**Time Spent Reviewing:**

6

---

> ### Author Response · Authors · 2021-08-06
> **Author's response**
>
> We thank the reviewer for the useful feedback, and are glad to hear their positive comments on the quality of the presentation and technical details of the paper.
>
> We address their concerns below:
> 1.  **Bypassing/addressing the drawbacks of bisimulation:** We agree with the reviewer's point, and will adjust the discussion here to give a more nuanced view on why the identified drawbacks for bisimulation do not apply to MICo.
> 1.  **Comparison b/w MICo and bisimulation:** This is a great point the reviewer raises, which we respond to below.
>      1.  **What does MICo lose out from ($\pi$-)bisimulation?** The main drawback of the MICo distance relative to bisimulation is in the tightness of the upper bound on the value function: the MICo distance, while still an upper bound on differences in value function, is much looser than bisimulation (compare the green and blue lines in Figure 3, left). The reduced MICo ($\Pi U^{\pi}$), although “closer” to the difference in value functions (compare orange and blue lines in Figure 3, left), is no longer an upper bound (as demonstrated in Lemma 5.1). We will make these downsides more explicit in the paper.
>      1.  **Visualizations on GridWorlds:** Generating these visualizations on GridWorlds to compare the different metrics is a good idea, and we will add them to the final version.
> 1.  **Explicit representations:** Could the reviewer please clarify their interpretation of explicitness in this context? We would then be happy to respond in the discussion period.
> 1.  **Non-stationarity:** The reviewer is correct in pointing out that the policy is changing throughout learning. Whether the learning process is improved by incorporating the metric loss during control learning is an empirical question, which our results demonstrate benefit from the addition of the MICo loss.
> 1.  **Typos:** Thank you, we will correct these!

---

> > ### Comment · Reviewer_dAxX · 2021-08-14
> > **Feedback to authors' response**
> >
> > Thank the authors for addressing my questions.
> >
> > Regarding the explicitness, I am referring to the authors' description in Line 85: "we learn this state similarity explicitly: that is, without the aid of any additional network parameters." I think it is better to rephrase it because whether or not using additional network parameters is more on the implementation side.

---

> > > ### Author Response · Authors · 2021-08-16
> > > **On explicitness comment**
> > >
> > > We thank the reviewer for the clarification. We will expand and rephrase this point in the final version, making clear that the use of extra network parameters in auxiliary losses is ultimately an implementation detail, and that the more fundamental notion is whether the agent's representation directly or indirectly features in the loss function.

---

### Official Review · Reviewer_PkRV · 2021-07-14

**Rating:** 7
**Confidence:** 4

**Summary:**

The paper proposes a new behavioral distance over the state space of an MDP, which can be used to learn representations in deep RL tasks. The proposed distance has three advantages over the existing bisimulation metric: (1) it has better computational complexity; (2) it allows for online approximation; and (3) it has a close connection to the underlying policy. Empirical evidence on toy examples shows the informativeness of the representations learned by using the proposed distance, and large-scale experiments show improvements of deep RL agents when incorporating the proposed distance to learn representations.


**Limitations And Societal Impact:**

The authors have adequately addressed the limitations and potential negative societal impact of their work.

**Main Review:**

### Strengths
- The paper is very relevant to the representation learning / (deep) RL community.
- The proposed MICo distance is a natural extension of the existing on-policy bisimulation metric, which removes the restriction to only deterministic MDPs. In addition, the theoretical analyses and proofs in Section 4 and Section 5 are novel to the best of my knowledge.
- I appreciate the paper's technicality level. The authors start by mentioning the inherent limitations of the bisimulation metric, propose the MICo distance, and then provide theoretical results to show how it overcomes these limitations. The results of the paper are significant, and the analyses and proofs are thorough and accurate.
- The paper is very well written. I enjoyed reading the paper.

### Weaknesses
- In my opinion, the lack of connection to the optimal policy is actually a disadvantage of MICo against the bisimulation metric. Imagine when the observation contains a lot of rich but task-irrelevant information, using the bisimulation metric will result in representations that are informative of the value similarities under the *optimal* policy. In other words, the resulting representations can successfully capture task-relevant information while filtering out redundant information.
- Using MICo as an additional loss only improves the performance of M-IQN and Rainbow. The results with and without MICo in other methods are fairly the same. Moreover, in some Atari games, using MICo makes the performance drop significantly. Why is this the case?
- While the authors have experimented MICo on many Atari games and with many different RL methods, the paper is lacking comparisons with other distances. I would like to see at least the comparison between MICo and the bisimulation metric used in [1].

### Other comments and questions
- Between lines 259 and 260, how did the authors come up with this specific way of parameterizing the MICo distance? There must be other parameterizations that are positive and symmetric.
- The first equation between lines 161 and 162, there should be a sum over all actions $(a,b)$.

I am willing to raise my score if these concerns are addressed.

[1] Zhang, Amy, et al. "Learning invariant representations for reinforcement learning without reconstruction." arXiv preprint arXiv:2006.10742 (2020).

**Time Spent Reviewing:**

16

---

> ### Author Response · Authors · 2021-08-06
> **Author's response**
>
> We thank the reviewer for the very useful feedback, and the positive comments on the theoretical results, writing quality, and relevance of the work to representation learning & deep RL!
>
> We will respond to the points raised below:
> 1.  **Lack of connection to optimal policies:** MICo, being a function of $\pi$, can in fact have a connection to $\pi^*$ (when $\pi = \pi^*$). The difficulty with traditional bisimulation metrics is that they _only_ have a connection to $\pi^*$ (by virtue of the maximization term in the update operator). Given that in standard RL the agent does not have access to $\pi^*$ a priori, and in large-scale environments is not likely to ever learn $\pi^*$, the use of bisimulation metrics is not informative of the policies the agent encounters _en route_ to $\pi^*$. Contrastingly, since MICo can be defined with respect to any policy $\pi$, it can yield useful representations for all policies throughout the learning process. Nevertheless, the general question that the reviewer raises - which policies are important to learn about in auxiliary tasks - is an important one, and one which we believe future work in the RL/representation learning communities will help to clarify; we will add further discussion of this point to the main paper.
> 1.  **Performance increase/decrease:** The aggregate plots in Figure 4 may make it appear that there are gains only for Rainbow and M-IQN, but they provide only a partial picture. Figures 6-11 in the appendix provide a more complete picture, where we can see that for all agents, MICo provides an improvement for a majority of games. It is true that MICo does not provide gains across all games, but we note that it is extremely rare for any new algorithm to provide gains across all 60 games. However, we hypothesize that the lack of improvements in some games is due to differing scales between the TD and MICo losses. An interesting avenue for future research is to investigate using an adaptive $\alpha$ (which mixes the TD and MICo losses), as opposed to a constant value like we used in the paper. We will add a discussion of this idea.
> 1.  **Comparison with [Zhang et al., 2020]:** The method of Zhang et al. requires Gaussian transitions (and a model estimate), as their formulation of the Wasserstein depends on this. Unfortunately, transitions in the ALE are in general not Gaussian, nor is the model available, so their method is unfortunately not applicable.
> 1.  **Choice of parameterization:** The parameterized form we use in L259 is not the only one possible to approximate a diffuse metric. We chose this particular form due to its connection to $\Pi U^{\pi}$, as discussed in lines 270--278. We will modify the discussion before L259 to clarify this connection.
> 1.  **Sum over all actions in L161-162:** Thank you for pointing this out, we will correct this.

---

> > ### Comment · Reviewer_PkRV · 2021-08-13
> > **Further discussion on comparison with bisimulation**
> >
> > I thank the authors for the detailed response. However, my concern regarding the comparison between MICO and bisimulation [Zhang et al., 2020] remains. While it is true that this baseline method only works with Gaussian transitions, I believe it's still useful to try to compare that with MICO. Can we simply learn a Gaussian dynamics and then use bisimulation to shape the latent space, similarly to what was done in [Zhang et al., 2020]?

---

> > > ### Author Response · Authors · 2021-08-16
> > > **Comparing to [Zhang et al., 2020]**
> > >
> > > We thank the reviewer for their further comments.
> > >
> > > While we agree this would be a useful comparison, learning a dynamics model on the ALE is quite challenging and has been the focus of entire research papers (see e.g. [Hafner et al., 2018](https://arxiv.org/abs/1811.04551); [Hafner et al., 2020](https://arxiv.org/abs/1912.01603); [Kaiser et al., 2020](https://arxiv.org/abs/1903.00374); [Hafner et al., 2021](https://arxiv.org/abs/2010.02193)). Each of these works use non-trivial encoder-decoder network architectures that go far beyond Gaussian transitions.
> > >
> > > What may be more feasible is to use our method on the DM-Control experiments used by [Zhang et al., 2020]. We have managed to get the code provided by [Zhang et al., 2020] running on our computers and are currently running some experiments to verify we can reproduce the results in their paper.
> > >
> > > Given that our implementation was in JAX, we will unfortunately have to reimplement our method in PyTorch (as that is what [Zhang et al., 2020] are using). We will do our best to obtain some results before the end of the discussion period, but in any case, we will add these extra results to the final version of the paper.

---

> > > > ### Comment · Reviewer_PkRV · 2021-08-19
> > > > **Comparing to [Zhang et al., 2020]**
> > > >
> > > > I thank the authors for their response,
> > > >
> > > > I do agree that learning a dynamics model on the ALE is challenging. However, in order to apply DBC, we only need to learn a dynamics model and a reward function **in the latent space**. This means we encode the current frames and future frames, and then learn a latent dynamics that maps the current latent vector to the future latent vector. I think this is also possible in MICO implementation? If yes I believe this can be done quicker than trying to make MICO work on DMControl tasks.

---

> > > > > ### Author Response · Authors · 2021-08-20
> > > > > **DBC on ALE**
> > > > >
> > > > > We thank the reviewer for their suggestion. Implementation-wise it should not be too difficult to add a dynamics model on the latent space, as the reviewer suggests. However, it may prove challenging to find the best hyperparameters for the dynamics model so as to make it a fair comparison against the MICo results.
> > > > >
> > > > > Nonetheless, we will do our best to implement DBC on our codebase, find a good set of hyperparameters for it, and report the results comparing to MICo.

---

> > > > > > ### Author Response · Authors · 2021-08-30
> > > > > > **Comparison to Zhang et al., 2020**
> > > > > >
> > > > > > We are happy to report that we have run comparisons to DBC. Please see the top-level comment we submitted with all the details.
> > > > > > Thank you for the suggestion!

---

> > > > > ### Comment · Reviewer_PkRV · 2021-08-31
> > > > > **Updated score**
> > > > >
> > > > > I thank the reviewer for their effort of running the additional comparisons with DBC.
> > > > >
> > > > > I am satisfied with the results, and believe MICO will have a great contribution to the RL/representation learning community. I'm raising my score accordingly.

---

### Official Review · Reviewer_apPQ · 2021-07-15

**Rating:** 7
**Confidence:** 3

**Summary:**

This work proposes a new formulation for quantifying the similarity of states based on rewards and next states. It points out three limitations of the bisimulation metric, and introduces the MICo distance. Given a specific policy, the MICo distance is defined with the MICo update operator, which the authors prove is a contraction mapping. They demonstrate some properties of it especially as a solution to the issues of the bisimulation metric. The authors suggest a specific form of approximation to the unique fixed point of the contraction mapping, so that it can be used for shaping state representations in environments with large state spaces. Empirically, the authors performed experiments to show the usefulness of a modified version of the MICo distance's learned approximation equipped with UMAP for predicting value functions and how the scores are improved with the state representations trained jointly with the MICo loss.

**Limitations And Societal Impact:**

I think the authors have fairly covered the limitations of their work.
Please see my main review, as well.

**Main Review:**

**Originality**

They suggest the new operator for defining the MICo distance. While it can be thought of as a simplified version of the bisimulation operator with the eliminated need for the Wasserstein metric, they describe how it differs from previous approaches, importantly addressing some issues of the bisimulation operator. Also, they suggest the learned approximation to the MICo distance for practical use.
I think the proposed methods and derivations are meaningfully novel.

**Quality**

I find the theory up to Sec. 4 (including the MICo distance with its operator $T_M^\pi$, equivalence to the Bellman operator in the auxiliary MDP, the online approximation of the solution, upper-bounding of the value function difference, etc.) technically sound.

In Sec. 5, I think the definition of $U_{\omega}$ (L259) is somewhat abrupt. Despite the usefulness of $\Pi U^\pi$ shown throughout Sec. 5 and its connection to $U_{\omega}$, I believe addressing why the parametrized approximation of $U^\pi$ should be (or at least can reasonably be) the form at L259 will be a good addition to the paper.

Regarding the experiments in Sec. 6, while it is extensive that the various RL algorithms with or without the MICo loss are evaluated on the large number of Atari environments, including some additional comparison with other baseline representation shaping methods would be great.

**Clarity**

The paper is overall easy to follow and well-organized.

Typo "oppoosed" at 328.

**Significance**

The theoretical results in this work, despite providing the diffuse metric not a metric, seem to have a good chance to be used in future literatures, thanks to the nice properties it has (bounding on an arbitrary policy's value function gaps, the approximation schemes, and the reduced computational complexity).

**Time Spent Reviewing:**

6

---

> ### Author Response · Authors · 2021-08-06
> **Author's response**
>
> We thank the reviewer for the useful comments, and are particularly happy that the reviewer found the central contributions of the paper novel and the paper well organised.
>
> We address the concerns mentioned below:
> 1. **Choice of parameterization for $U^\pi$:** The parameterized form we use in L259 is not the only one possible to approximate a diffuse metric. However, as the reviewer points out, we chose this form due to its connection to $\Pi U^{\pi}$. We will modify the discussion before L259 to clarify this connection.
> 1. **Including additional comparisons:** The natural comparisons would be the $\pi$-bisimulation metric of [[Castro, 2020]](https://arxiv.org/abs/1911.09291) and [[Zhang et al., 2021]](https://openreview.net/pdf?id=-2FCwDKRREu), but these require specific assumptions on the transition dynamics (deterministic and Gaussian transitions, respectively) which are not applicable in the Atari domains we study here. Time permitting, we can try to run some experiments before the discussion period is over if the reviewer has particular representation learning methods in mind, but as described above and in the paper, the most relevant methods are not applicable in this setting.
> 1.  **Typo:** Thank you for pointing out the typo, we will correct it.

---

> > ### Comment · Reviewer_apPQ · 2021-08-18
> > **Response to the authors**
> >
> > I thank the authors for the response.
> >
> > I appreciate that the authors are trying to compare their method with [Zhang et al., 2020] on DM-Control, as suggested by one of their other comments.
> >
> > Regarding the parameterization for $U^\pi$, I still think it would be valuable to add some reasoning for it without respect to $\Pi U^\pi$, because while $\Pi U^\pi$ is examined in Sec. 5, the same parameterization is used in Sec. 6 for more empirical evaluations but independently of $\Pi U^\pi$.

---

> > > ### Author Response · Authors · 2021-08-19
> > > **On the chosen parameterization**
> > >
> > > We thank the reviewer for the suggestion.
> > >
> > > The chosen parametrization the reviewer mentions allows us to express non-zero self-distances, a key property of the MICo distance. The reduced MICo distance $\Pi U^{\pi}$ is in fact crucial to the empirical evaluations: the representations $\phi_\omega$, when used for value estimation, exist in a Euclidean space and thus in this case their distance is a proper (pseudo-)metric (e.g. self-distances are always zero). The reduction $\Pi U^{\pi}$ is thus necessary to reconcile for the fact that the MICo loss, which directly shapes $\phi_\omega$, is used to learn a _diffuse_ metric, with non-zero self-distances.
> > >
> > > However, we agree that it can be better motivated if we introduce the exact parameterization after discussing the necessity of the reduction, and will do so for the final version.

---

### Official Review · Reviewer_vaui · 2021-07-17

**Rating:** 7
**Confidence:** 3

**Summary:**

The paper presents MICo - a behavioural metric to learn representations for deep RL agents and shows its effectiveness - both theoretically and empirically (on the Arcade Learning Environment benchmark).

**Limitations And Societal Impact:**

Limitations And Societal Impact are addressed.

**Main Review:**

## Strengths

1. The paper tackles an interesting, relevant and impactful problem in RL.
2. The paper is very well written. I am not much familiar with the convergence proof techniques etc but I did not have much trouble understanding the paper.
3. The paper provides both theoretical and empirical results. The empirical results show that MICo can be combined with a large number of baselines.

## Suggestions for improvement

1. Figure 4 is very hard to parse. Maybe the lines could be thinner and "...."can be used in place of "----". Alternatively, consider moving some of the results to the appendix. Also consider adding error bars to show the variance.
2. Why arent other bisimulation metric based methods reported in the experiment section? I understand they have additional assumptions baked in (line 82) but some of them should still be evaluated and reported results with.


An additional comment (Note that I am not holding this as a weakness for the paper): If the authors have resources, they should consider reporting the results over at-least 10 seeds. They should also consider releasing the metrics as a csv/json file. I expect this paper to be well-cited and several other future works to use this as a baseline.

**Time Spent Reviewing:**

3

---

> ### Author Response · Authors · 2021-08-06
> **Author's response**
>
> We thank the reviewer for the useful comments, and are glad to hear their positive comments on the importance of the problem studied, and the quality of the writing in the paper.
>
> Below we respond to further comments made.
> 1.  **Figure 4:** We will incorporate the suggestions to improve legibility on Figure 4.
> 1.  **Other bisimulation metrics:** Other bisimulation metrics are not directly applicable to the Atari environments considered in this paper, as the reviewer mentions: (i) π-bisimulation requires deterministic transitions, (ii) the bisimulation metric requires solution of intractable optimal transport problems to compute, (iii) the version of the bisimulation metric considered by [Zhang et al. (2021)](https://openreview.net/pdf?id=-2FCwDKRREu) requires known Gaussian transitions. While we agree it would be an interesting research question to understand whether these definitions can be extended to tractably apply to environments such as Atari games, we would argue that this is non-trivial (but an excellent topic for future work), since rigorously modifying bisimulation to be applicable in general stochastic environments is the core contribution of this paper.
> 1.  **Number of seeds:** We appreciate the reviewers point regarding number of seeds. Given that we ran experiments over the 60 games for 5 agents, we felt 5 seeds (totalling 1500 separate runs) was a good balance between statistical significance and computational/environmental expense. In addition, we include bootstrap CIs in Figures 6-11 to indicate the extent to which the differences in performance for these agents are statistically significant.
> 1.  **JSON files:** Releasing JSON files is a great idea, and we will include them in our open-source release.

---

> > ### Comment · Reviewer_vaui · 2021-08-18
> > **Thank you for the response**
> >
> > I thank the authors for their response and I am happy that they will be releasing the JSON files for the results.

---

### Author Response · Authors · 2021-08-30
**Comparison to Zhang et al., 2021**

We have run a number of experiments comparing MICo against DBC, the method in [Zhang et al., 2021](https://arxiv.org/abs/2006.10742). As detailed below, our experiments indicate that MICo can yield benefits when added to SAC and/or DBC, demonstrating its flexibility and practicality.

We found these results quite positive and encouraging, and would thus like to thank the reviewers for suggesting them. We will add these results and a discussion in the paper.

## Experimental details

The Deep Bisimulation for Control (DBC) method introduced by Zhang et al. consists of adding to SAC:
-  a reward model (on the latent states)
-  a dynamics model (on the latent states)
-  a bisimulation loss on the latent states (equation (4) in their paper).

To incorporate MICo into SAC, we applied the MICo loss (defined on the bottom of page 6 in our paper) on the same latent states used by DBC.

In addition to comparing SAC, DBC, and MICo, we added the following comparisons:
-  **DBC-Det**: Although the DBC agent is described using a stochastic transition model, we noticed in [the code for DBC](https://github.com/facebookresearch/deep_bisim4control) that the dynamics model [defaulted to a deterministic dynamics model](https://github.com/facebookresearch/deep_bisim4control/blob/master/transition_model.py#L94). DBC-Det is thus a version of DBC with this deterministic transition model.
-  **DBC+MICo**: We replaced the bisimulation loss of DBC (eqn (4) in Zhang et al.) with our MICo loss. It is worth noting that although the reward and dynamics losses of DBC are still being optimized, the MICo loss does _not_ depend on the dynamics model, as opposed to the bisimulation loss of Zhang et al., which does.

## Results

We ran on the same environments reported in Figure 3 of Zhang et al. (cheetah_run, walker_walk, finger_spin) on the [deepmind-control suite](https://github.com/deepmind/dm_control), running from pixels. The table below summarizes our results after 1 million environment stetps. Each agent was run with 5 independent seeds and we report the 75% confidence intervals in parentheses:

| **Env**  |  SAC  |  DBC  |   DBC-Det  |   DBC+MICo  |  MICo  |
|-------------|------|------|------|------|------|
|  cheetah_run |  157.16 (54.10, 260.22) | 102.26  (66.67, 137.85)  |  62.72  (43.68, 81.76)  |  148.29  (109.25, 187.34) |  **306.9 (267.02, 346.78)** |
| walker_walk | 48.04 (39.05, 57.05) | 88.22 (74.59, 101.86) | 71.83 (57.65, 86.02) | **169.84 (27.60, 312.08)** | 28.37 (17.13, 39.61) |
| finger_spin | 218.55 (185.81, 251.29) | 34.80 (24.33, 45.27) | 60.95 (35.79, 86.11) | **373.90 (350.99, 396.81)** | 224.05 (205.10, 243.00) |

As can be seen, in all three environments we have the best performance from either MICo or DBC+MICo. Given that no hyperparameter tuning was performed (other than to find a reasonable value for the $\alpha$ (see line 267 in our paper), which was $10^{-5}$), these results suggest that MICo can be a valuable addition either on its own, or in combination with any other method that uses a bisimulation-like loss.

We also ran experiments with the [distracting control suite](https://github.com/google-research/google-research/tree/master/distracting_control), which adds video distractors to the environments from the deepmind-control suite, in addition to adding "easy", "medium", and "hard" difficulty levels. We used the same settings as for the experiments above; the results below are reported after 90 thousand environment steps, as the distracting-control suite was substantially slower to run. Due to computational constraints we were not able to run the DBC-Det in this setting.


| **Env**  |  Difficulty | SAC  |  DBC  |   DBC+MICo  |  MICo  |
|-------------|------|------|------|------|------|
| cheetah_run | easy | **23.51	(19.23, 27.79)** | 3.95	(2.38, 5.53) | 3.99	(2.83, 5.15) | **27.94	(23.13, 32.75)** |
| | medium | **17.89	(15.44, 20.34)** | 4.66	(3.21, 6.12) | 5.72	(4.88, 6.56) | **15.07	(8.42, 21.73)** |
| | hard | **13.90	(5.16, 22.64)** | 3.80	(2.25, 5.35) | 5.35	(4.15, 6.54) | **19.88	(15.89, 23.87)** |
| walker_walk | easy | 12.96	(10.51, 15.41) | 13.18	(9.54, 16.82) | 14.57	(12.97, 16.17) | 13.09	(10.82, 15.37) |
| | medium | 13.04	(7.60, 18.48) | 10.36	(9.59, 11.14) | 14.45	(12.41, 16.49) | 11.36	(8.21, 14.52) |
| | hard | 12.04	(10.26, 13.81) | 11.62	(9.91, 13.33) | 13.87	(10.78, 16.97) | 11.34	(8.43, 14.24) |
| finger_spin | easy | **12.25	(-0.63, 25.13)** | 0.00	(0.00, 0.00) | 5.85	(1.27, 10.43) | **34.65	(6.48, 62.82)** |
| | medium | 2.50	(0.11, 4.89) | 0.20	(-0.15, 0.55) | 4.25	(0.72, 7.78) | 2.80	(-0.71, 6.31) |
| | hard | 3.30	(0.77, 5.83) | 0.55	(-0.07, 1.17) | 4.40	(0.61, 8.19) | 0.50	(-0.37, 1.37) |

This suite proved quite challenging for all agents. However, we can see that although in *cheetah_run* adding the DBC components to SAC are quite detrimental to performance, MICo either maintains or improves performance; a similar conclusion can be reached in *finger_spin* easy. In the other settings none of the agents were able to make much progress.


## Some implementation details
### Framework used
In order to be able to re-use our MICo code in a new setting, we decided to use an existing JAX version of SAC (as described by [Haarnoja et al, 2019](https://arxiv.org/abs/1812.05905)) and incorporate our MICo loss directly on it. We implemented the losses and network architectures of DBC by following the code provided by the authors. We include the new SAC code with the rest of the code which we will release with the paper.

### Distractors
The distractor videos used by the DBC authors seems to have come from the [dmc2gym](https://github.com/denisyarats/dmc2gym) wrapper. However, we were unable to use this wrapper for distractors, as the authors appear to have used a different version than the one currently available on GitHub. In particular, the distractors are specified via the [`img_source`](https://github.com/facebookresearch/deep_bisim4control/blob/master/train.py#L36) flag, which are [passed](https://github.com/facebookresearch/deep_bisim4control/blob/master/train.py#L295) to the [`make`](https://github.com/denisyarats/dmc2gym/blob/master/dmc2gym/__init__.py#L5) function of `dmc2gym`. As can be seen, this `make` function no longer accepts an `img_source` parameter.

Thus, we decided to use the [distracting-control suite](https://github.com/google-research/google-research/tree/master/distracting_control), which achieves a similar result (adding distracting video backdrops to the dm_control foreground).

---

### Decision · Program_Chairs · 2021-09-27

**Decision:**

Accept (Poster)

**Comment:**

This seems like a solid paper that develops new approach to the interesting problem of inducing bias in learning through a notion of state similarity.  The notion overcomes some limitations of previous notions that were based on bisimulation concepts.  The reviewers are aligned in their ratings of the paper.